# Enhancing Human Experience in Human-Agent Collaboration: A Human-Centered Modeling Approach Based on Positive Human Gain

Yiming Gao[1*]  Feiyu Liu[1*]  Liang Wang[1]  Dehua Zheng[1]  Zhenjie Lian[1]
Weixuan Wang[1]  Wenjin Yang[1]  Siqin Li[1]  Xianliang Wang[1]  Wenhui Chen[1]
Jing Dai[1]  Qiang Fu[1]  Wei Yang[1]  Lanxiao Huang[2]  Wei Liu[1]
[1]Tencent AI Lab, Shenzhen, China   [2]Tencent TiMi L1 Studio, Chengdu, China
`yatminggao@tencent.com;wl2223@columbia.edu`

## Abstract

Existing game AI research mainly focuses on enhancing agents' abilities to win games, but this does not inherently make humans have a better experience when collaborating with these agents. For example, agents may dominate the collaboration and exhibit unintended or detrimental behaviors, leading to poor experiences for their human partners. In other words, most game AI agents are modeled in a "self-centered" manner. In this paper, we propose a "human-centered" modeling scheme for collaborative agents that aims to enhance the experience of humans. Specifically, we model the experience of humans as the goals they expect to achieve during the task. We expect that agents should learn to enhance the extent to which humans achieve these goals while maintaining agents' original abilities (e.g., winning games). To achieve this, we propose the Reinforcement Learning from Human Gain (RLHG) approach. The RLHG approach introduces a "baseline", which corresponds to the extent to which humans primitively achieve their goals, and encourages agents to learn behaviors that can effectively enhance humans in achieving their goals better. We evaluate the RLHG agent in the popular Multi-player Online Battle Arena (MOBA) game, *Honor of Kings*, by conducting real-world human-agent tests. Both objective performance and subjective preference results show that the RLHG agent provides participants better gaming experience.

## 1 Introduction

Recently, Reinforcement Learning (RL) has been widely used in developing Artificial Intelligence (AI) systems for games, developing various agents that perform at a human-level performance, such as AlphaGo (Silver et al., 2016; 2017) in *Go*, AlphaStar (Vinyals et al., 2019) in *StarCraftII*, OpenAI Five (OpenAI et al., 2019) in *Dota2*, and Wukong AI (Ye et al., 2020a) in *Honor of Kings*. To further expand the applications of these agents, researchers are exploring ways to improve their generalization to human partners (Carroll et al., 2019; Hu et al., 2020; Strouse et al., 2021; Lupu et al., 2021; Knott et al., 2021; McKee et al., 2022; Yu et al., 2022), as well as enabling effective explicit communication between agents and humans (FAIR et al., 2022; Gao et al., 2022) in collaborative tasks. However, these agents primarily focus on maximizing their own rewards to complete the task, less considering the role of their human partners. This potentially leads to behaviors that are inconsistent with human values and preferences, resulting in a poor experience for human partners (Fisac et al., 2020; Alizadeh Alamdari et al., 2022). Thus, we say that the optimization objective of these agents is *self-centered*. In a qualitative study (Cerny, 2015) on companion behavior, it was found that humans reported greater enjoyment of the game when the AI assisted them more like a sidekick.

Consider the scenario depicted in Figure 1($\Leftarrow$), the self-centered agent may push the obstacle to the human side and pass through to get the coin itself, which can complete the task, but its human partner has no experience. However, as illustrated in Figure 1($\Rightarrow$), the human may prefer the agent to play a more assisting role by pulling the obstacle to its side, thereby facilitating the human to get the coin. Consequently, not only is the task completed, but the experience of the human, hereafter referred to as *human experience*, is also enhanced. In many real-world collaborative scenarios, such as robotic

---

*These authors contributed equally to this work.

assistants and autonomous driving, humans not only want to complete the task but also pursue a better experience (Wilson & Daugherty, 2018a;b; Crandall et al., 2018). Therefore, it is important for a collaborative agent to learn a *human-centered* objective, that is, to enhance the human experience.

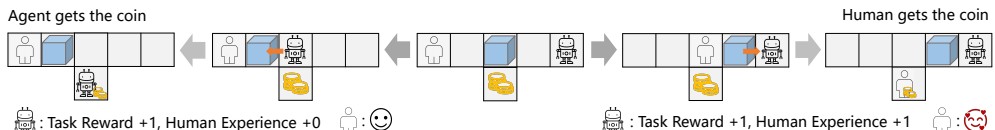

Figure 1: Toy scenario, where an agent and its human partner are on either side of an obstacle. Only the agent is capable of pushing or pulling the obstacle. Their task goal is to obtain the coin. ⇐: The agent gets the coin by itself. The task is completed, but the human has no experience. ⇒: The agent assists the human to get the coin. Both the task is completed and the experience of the human is enhanced.

In this paper, we conceptualize the human experience as the *human goals* they expect to achieve during the task. Note that the human experience is task-dependent, and the corresponding goals may be explicit, such as obtaining coins, improving safety, and enhancing *empowerment* (Du et al., 2020). Additionally, these goals may be implicit and require inference through goal reward models learned with human feedback (Ng et al., 2000; Ziebart et al., 2008; Ho & Ermon, 2016; Christiano et al., 2017; Ouyang et al., 2022). These human goals can then be quantified as human goal rewards, hereafter referred to as *human rewards*, which measure the extent to which humans achieve these goals. Our work does not aim to define or infer human goals accurately, but rather focuses on training agents to enhance the extent to which humans achieve these goals. One intuitive way is to directly combine agents' original (task-related) rewards with human rewards. However, we find that this approach may encounter the human-agent credit assignment challenge, where the human rewards are assigned to the agent without any assistive behavior, potentially leading to the agent learning incorrect behavior and even losing its *autonomy*, i.e., the original abilities to complete the task, as shown in Section 4.2. A potential solution to solve this is to carefully reshape the human reward function. However, this approach heavily relies on domain knowledge and expertise.

We propose a novel approach that enables agents to learn to enhance the extent to which humans achieve their goals while maintaining agents' autonomy as much as possible. Our key insight is that the contribution made by humans themselves needs to be separated from the human rewards, and the remaining benefits can be considered as the real contribution of the agents to enhancing the humans, which we refer to as *human gains*. To realize this insight, we propose the Reinforcement Learning from Human Gain (RLHG) approach, which involves two stages. Firstly, we evaluate the primitive human performance in achieving human goals and consider this as a "baseline". We train a value network to estimate the primitive expected human return in achieving human goals with episodes collected by directly teaming the human and the self-centered agent to execute. Secondly, we train the agent to learn effective human enhancement behaviors. We train a gain network to estimate the expected positive gain of human return when subjected to effective enhancement, compared to the "baseline". The agent is fine-tuned using the combination of its original advantage and the human-centered advantage calculated by the positive human gains.

We conducted experiments to evaluate the effectiveness of the RLHG approach in *Honor of Kings* (Wei et al., 2022), a typical Multi-player Online Battle Arena (MOBA) game (Silva & Chaimowicz, 2017), which has received much attention from researchers lately (Ye et al., 2020a;b;c; Gao et al., 2021; 2022). We first evaluated the RLHG approach in simulated environments, i.e., human model-agent tests. Our experimental results demonstrate that the RLHG agent outperforms baseline agents in enhancing the performance of the human model in achieving human goals. We further conducted real-world human-agent tests, with the RLHG agent teaming up with participants of varying skill levels. Our experimental results show that the RLHG agent effectively enhanced the performance of general-level participants in achieving their goals, bringing them closer to the performance of high-level participants. And we find that this enhancement is generalizable across different participant skill levels. Additionally, subjective preference results reveal that most participants were satisfied with the RLHG agent, believing it provided a better gaming experience. In general, our contributions are as follows:

- We proposed a human-centered modeling scheme for guiding human-level agents to enhance the experience of their human partners in collaborative tasks.

- We gained insights into the challenge of human-agent credit assignment and addressed this challenge by presenting the RLHG algorithm, along with a detailed implementation framework.

- We conducted human-agent tests in *Honor of Kings*, and both objective performance and subjective preference results show that the RLHG agent provides participants better gaming experience.

## 2 BACKGROUND

### 2.1 GAME INTRODUCTION

*Honor of Kings* is a typical MOBA game, characterized by multi-agent cooperation and competition mechanisms, often used as a testbed for game AI research (Wu, 2019; Ye et al., 2020a;b;c; Gao et al., 2021; 2022). The game is played by two opposing teams on a symmetrical map, each comprising five players. The game environment, depicted in Figure 2(a), consists of the main hero with peculiar skill mechanisms and attributes, controlled by each player. Players can maneuver the hero's movement using the wheel (C.1) and release the hero's skills through the buttons (C.2, C.3). They can view the local environment on the screen, the global environment on the mini-map (A), and access game states on the dashboard (B). The agent and the human player share the same game information and the action mechanism. Agents team up with human players to compete against the enemy camp through collaboration. The gaming experience is crucial for a player's engagement and satisfaction. Along with the task goal, players also pursue multiple individual goals (Figure 2(b)), such as achieving a higher MVP score, experiencing more highlight moments, and obtaining more in-game resources. The pursuit of these goals can contribute to a more enjoyable and rewarding gaming experience. Agents can enhance the gaming experience of their human partners through interactive behaviors, such as timely support, sharing resources, and active protection.

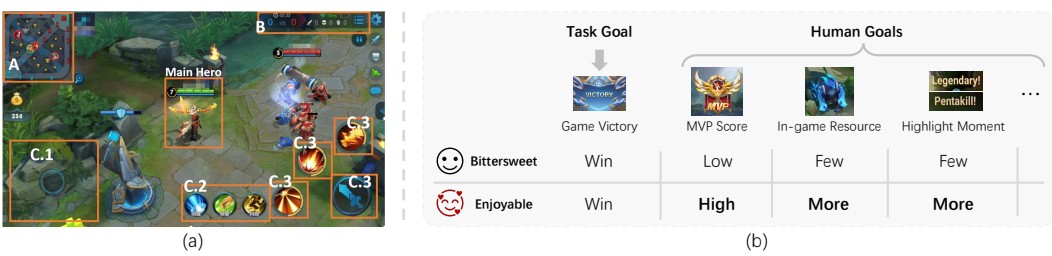

Figure 2: **(a)** The UI of *Honor of Kings*. **(b)** In-game goals, based on our participant survey (see Figure 4(c)). Human players pursue multiple goals for more enjoyable gaming experience.

### 2.2 HUMAN-AGENT COLLABORATION

We formalize a human-agent collaboration task as an extension of the Decentralized Partially Observable Markov Decision Processes (Dec-POMDP) (Bernstein et al., 2002), which can be represented as a tuple $< N, \mathbf{S}, \mathbf{A}, \mathbf{O}, P, R, \gamma, \pi_H, \mathcal{G}^H, R^H >$, where $N$ denotes the number of agents. $\mathbf{S}$ denotes the space of global states. $\mathbf{A} = \{\mathcal{A}^i, \mathcal{A}^H\}_{i=1,...,N}$ denotes the space of actions of $N$ agents and a human to be enhanced, respectively. $\mathbf{O} = \{\mathcal{O}^i, \mathcal{O}^H\}_{i=1,...,N}$ denotes the space of observations of $N$ agents and the human, respectively. $P : \mathbf{S} \times \mathbf{A} \rightarrow \mathbf{S}$ and $R : \mathbf{S} \times \mathbf{A} \rightarrow \mathbb{R}$ denote the shared state transition probability function and the task reward function of $N$ agents, respectively. $\gamma \in [0, 1)$ denotes the discount factor. We define an agent policy, $\pi^i$, to be a mapping from histories of observations $o^i = \{o^i_1, ..., o^i_t\} \in \mathcal{O}^i$ to actions $a^i \in \mathcal{A}^i$. A joint policy, $\pi_\theta = < \pi^1, ..., \pi^N >$, parameterized by $\theta$, is defined to be a tuple of $N$ agent policies. We define a human policy $\pi_H$, to be a mapping from observations $o^H \in \mathcal{O}^H$ to actions $a^H \in \mathcal{A}^H$, which remains fixed during the agent's optimization process and cannot be accessible to the agent. $\mathcal{G}^H = \{g_i\}_{i=1,...,M}$ denotes the human goals, where $g_i$ is a designated goal and $M$ is the total number of human goals. $R^H : \mathbf{S} \times \mathbf{A} \times \mathcal{G}^H \rightarrow \mathbb{R}$ denotes the human reward function.

Most previous research work focuses on learning self-centered agents. In agent-only team settings, the agent is optimized to complete the task, with the optimization objective being to maximize the value function for a given state $s$, i.e., $V^{\pi_\theta}(s) = \mathbb{E}_{\pi_\theta}[G_t|s_t = s]$, where $G_t = \sum_{k=0}^{\infty} \gamma^k R_{t+k+1}$ represents the discounted cumulative task rewards. In human-agent team settings, the agent is

optimized to adapt to its human partners to complete the task, and the corresponding optimization objective is $V^{\pi_\theta, \pi_H}(s) = \mathbb{E}_{\pi_\theta, \pi_H}[G_t | s_t = s]$. This optimization objective paradigm is referred to as the self-centered objective. While optimizing the self-centered objective may enhance the agents' abilities to complete tasks (i.e., task rewards $R$) in collaboration with humans, it does not necessarily enhance the human experience (i.e., human rewards $R^H$).

## 3 METHODS

In this section, we introduce the human-centered modeling scheme. We start with formalizing the human-centered objective (Section 3.1). Then we propose a novel insight that enables agents learn to enhance the extent to which humans achieve their goals while maintaining the agents' autonomy (Section 3.2). Finally, we implement our insights by providing the RLHG algorithm and a practical implementation (Section 3.3).

### 3.1 HUMAN-CENTERED OBJECTIVE

We define the human-centered objective as $V_H^{\pi_\theta, \pi_H}(s) = \mathbb{E}_{\pi_\theta, \pi_H}[G_t^H | s_t = s]$, where $G_t^H = \sum_{k=0}^{\infty} \gamma^k R_{t+k+1}^H$ is the discounted cumulative human rewards. Intuitively, the self-centered objective optimizes the agents' abilities to complete the task, while the human-centered objective optimizes the agents' abilities to enhance human performance in achieving human goals $\mathcal{G}^H$. Thus, the overall optimization objective can be formulated as:

$$J(\theta) = V^{\pi_\theta, \pi_H}(s) + \alpha \cdot V_H^{\pi_\theta, \pi_H}(s) = \mathbb{E}_{\pi_\theta, \pi_H}\left[G_t + \alpha \cdot G_t^H | s_t = s\right], \tag{1}$$

where $\alpha$ is a balancing parameter. The approximation to the policy gradient is defined as follows:

$$\nabla J(\theta) = \mathbb{E}_{\pi_\theta, \pi_H}\left[\sum_{t=0}^{\infty} \nabla_\theta \log \pi_\theta(a_t | o_t)\left(G_t + \alpha \cdot G_t^H\right)\right], \tag{2}$$

$$\propto \mathbb{E}_{\pi_\theta, \pi_H}\left[\sum_{t=0}^{\infty} \nabla_\theta \log \pi_\theta(a_t | o_t)\left(A(s_t, a_t) + \alpha \cdot A_H(s_t, a_t)\right)\right], \tag{3}$$

where $A(s, a) = \mathbb{E}_{\pi_\theta, \pi_H}[G_t | s_t = s, a_t = a] - V^{\pi_\theta, \pi_H}(s)$ is the self-centered advantage and $A_H(s, a) = \mathbb{E}_{\pi_\theta, \pi_H}[G_t^H | s_t = s, a_t = a] - V_H^{\pi_\theta, \pi_H}(s)$ is the human-centered advantage.

However, incorporating human rewards directly into the optimization objective may lead to negative consequences, such as human-agent credit assignment issues. Intrinsically, humans possess the primitive abilities to achieve certain goals independently. Therefore, it is unnecessary to reward the agents for assisting when the human can easily achieve human goals or to reward the agents who do not provide any assistance, as it potentially leads to the agents learning incorrect behavior and even losing their autonomy. To this end, we propose a novel insight below to achieve effective human enhancement without compromising the agents' original abilities to complete the task.

### 3.2 EFFECTIVE HUMAN ENHANCEMENT

Our key insight is that the human contribution, termed *human primitive value*, should be distinguished from human rewards. The residual benefits, representing the agent's actual contribution in enhancing human to achieve goals, are referred to as *human gains*.

We define the *human primitive value* as $V_H^{\pi_0, \pi_H}(s)$, which represents the expected human return for a given state $s$ under the setting of collaboration between the human $\pi_H$ and the pre-trained agent $\pi_0$. We define the *human gain* as $\Delta(s, a) = \mathbb{E}_{\pi_\theta, \pi_H}[G_t^H | s_t = s, a_t = a] - V_H^{\pi_0, \pi_H}(s)$, which represents the benefits brought by taking a specific action $a$ for a given state $s$ compared to the *human primitive value*. In the process of learning to enhance the human, the agents explore two types of behaviors in state $s$: effective enhancement behaviors, i.e., $A^+(s) = \{a | a \sim \pi_\theta, \Delta(s, a) > 0\}$, and invalid enhancement behaviors, i.e., $A^-(s) = A(s) \setminus A^+(s) = \{a | a \sim \pi_\theta, \Delta(s, a) \leq 0\}$. Intuitively, $A^+$ can help the human achieve human goals better than the primitive and $A^-$ provides no benefits or even hinders the human from achieving human goals. Therefore, the agent is only encouraged to learn effective enhancement behaviors, and the Eq. 3 can be reformulated as:

$$\nabla J(\theta) = \mathbb{E}_{\pi_\theta, \pi_H} \left[ \sum_{t=0}^{\infty} \nabla_\theta \log \pi_\theta(a_t|o_t) \left( A(s_t, a_t) + \alpha \cdot \widehat{A}_H(s_t, a_t) \right) \right], \tag{4}$$

where $\widehat{A}_H(s, a) = \Delta(s, a) - \widehat{\Delta}(s)$ is the advantage of the human gain over the expected *positive human gain* $\widehat{\Delta}(s) = \mathbb{E}_{a \sim A^+(s)}[\Delta(s, a)]$. We use $\Delta_\omega(s)$ to denote an estimate of $\widehat{\Delta}(s)$, which can be trained by minimizing the following loss function:

$$L(\omega) = \mathbb{E}_{s \in S, a \in A} \left[ \mathbb{I}(\Delta(s, a)) \cdot \|\Delta(s, a) - \Delta_\omega(s)\|_2 \right], \quad \mathbb{I}(x) = \begin{cases} 1, & x > 0 \\ 0, & x \leq 0 \end{cases} \tag{5}$$

where $\mathbb{I}$ in Eq. 5 is an indicator function to filter invalid enhancement samples.

### 3.3 THE ALGORITHM & PRACTICAL IMPLEMENTATION

---
**Algorithm 1** Reinforcement Learning from Human Gain (RLHG)

---
1: **while** not converged **do**
2:     Freeze $\pi_0$ and collect human-agent team $< \pi_0, \pi_H >$ trajectories;
3:     Compute human return $G^H$, and update human primitive value $V_\phi \leftarrow G^H$;
4: **end while**                    // Stage I: Human Primitive Value Estimation
5: **while** not converged **do**
6:     Freeze $V_\phi$ and collect human-agent team $< \pi_\theta, \pi_H >$ trajectories;
7:     Compute original return $G$, self-centered advantage $A$;
8:     Compute human return $G^H$, human gain $\Delta$, and human-centered advantage $\widehat{A}_H$;
9:     Update agent policy network $\pi_\theta$ using Eq. 4, and value network $V_\psi \leftarrow G$;
10:     Update human gain network $\Delta_\omega$ using Eq. 5;
11: **end while**                    // Stage II: Human Enhancement Training

---

We introduce the RLHG algorithm 1 to actualize our insights, comprising two stages: the Human Primitive Value Estimation (Stage I) and the Human Enhancement Training (Stage II). In Stage I, the RLHG algorithm freezes the pre-trained agent policy $\pi_0$ and collects trajectory samples to compute the human return $G^H$. Subsequently, the RLHG algorithm updates the human primitive value network $V_\phi$ by minimizing the Temporal Difference (TD) errors (Sutton & Barto, 2018). In Stage II, the RLHG algorithm freezes $V_\phi$ and collects trajectory samples to compute the original return $G$ and human return $G^H$. Both $G$ and $G^H$ contribute to determining the self-centered advantage $A$ and human-centered advantage $\widehat{A}_H$, respectively. The agent's policy network $\pi_\theta$ is fine-tuned according to Eq. 4 and the value network $V_\psi$ is fine-tuned by minimizing the TD errors. $\Delta_\omega$ is trained by minimizing the loss function in Eq. 5.

Figures 3(a) and (b) show the training framework of Stage I and Stage II, respectively. The human policy $\pi_H$ can be trained via Behavior Cloning (BC) (Bain & Sammut, 1995) or any Supervised Learning (SL) techniques (Ye et al., 2020b), but this is not the focus of our concern. Since in many practical scenarios, agents cannot access the human policy, we instead learn a human policy embedding, enabling agents to adapt and infer human intentions. Similar to the Theory-of-Mind (Rabinowitz et al., 2018), we input the observed human historical information $h_t = (s_t^H, ..., s_1^H)$ into an LSTM module (Hochreiter & Schmidhuber, 1997) to extract the human policy embedding. The human policy embedding is subsequently fed into two extra value networks, i.e., the human primitive value network $V_\phi$ and the gain network $\Delta_\omega$, and fused into the agent's original network $\pi_\theta$. We use *surgery* techniques (Chen et al., 2015; OpenAI et al., 2019) to fuse the human policy embedding into the agent's original network, i.e. adding more randomly initialized units to an internal fully-connected layer. We apply the absolute activation function to ensure that the predicted gains are non-negative. In practical training, we find that only conducting human enhancement training has a certain negative impact on the agent's original abilities to complete the task. Therefore, we introduce $1 - \beta\%$ agent-only team settings to maintain the agent's original abilities and reserve $\beta\%$ human-agent team settings to learn effective enhancement behaviors. These two environments can be easily controlled through the task gate, i.e., the task gate is set to 1 in the human-agent team settings and 0 otherwise.

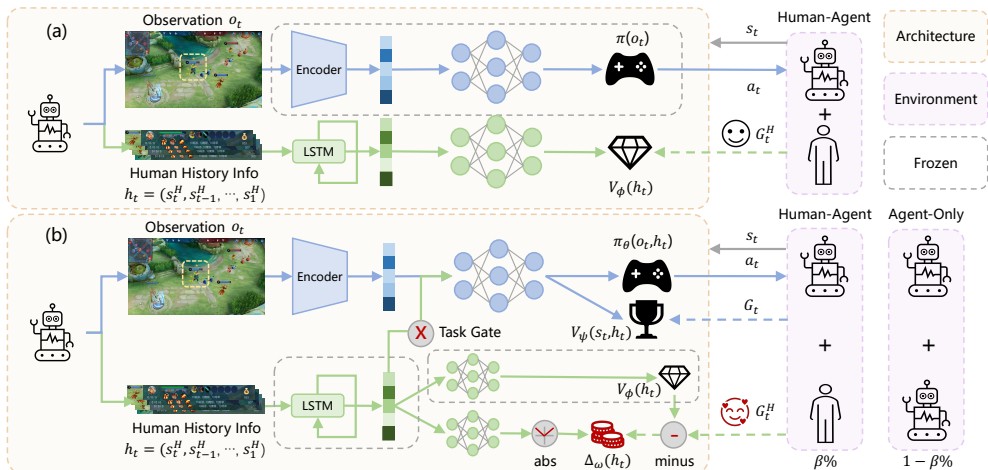

Figure 3: **The RLHG training framework.** (a) Human Primitive Value Estimation stage. The human primitive value network $V_\phi$ is trained in the human-agent team settings with the agent's policy $\pi$ frozen. (b) Human Enhancement Training stage. $V_\phi$ is frozen and added to a downstream network $\Delta_\omega$ to learn to estimate the expected positive human gain. $\beta\%$ human-agent team settings are used to learn human enhancement behaviors, and $1 - \beta\%$ agent-only team settings are used to maintain the agent's original ability.

# 4 EXPERIMENTS

In this section, we evaluate the proposed RLHG approach by conducting both simulated human model-agent tests and real-world human-agent tests in *Honor of Kings* (HoK). All experiments[1] were conducted in the 5v5 mode with a full hero pool (over 100 heroes, see Appendix A.2). Our demo videos and code are present at https://sites.google.com/view/rlhg-demo.

## 4.1 EXPERIMENTAL SETUP

**Environment Setup:** To evaluate the performance of the RLHG agent, we conducted experiments in both the simulated environment, i.e., human model-agent game tests, and the real-world environment, i.e., human-agent game tests, as shown in Figure 4(a) and 4(b), respectively. All game tests were played in a 5v5 mode, that is, 4 agents plus 1 human or human model team up against a fixed opponent team. To conduct our experiments, we communicated with the game provider and obtained testing authorization. The game provider assisted in recruiting 30 experienced participants with anonymized personal information, which comprised 15 high-level (top 1%) and 15 general-level (top 30%) participants. We first did an IRB-approved participant survey on what top 5 goals participants want to achieve in-game, and the result is shown in Figure 4(c). We can see that the top 5 goals voted the most by the 30 participants including the task goal, i.e., game victory, and 4 human goals, i.e., high MVP score, high participation, more highlights, and more resources. We find that participants consistently rated the high MVP score goal most, even more than the task goal.

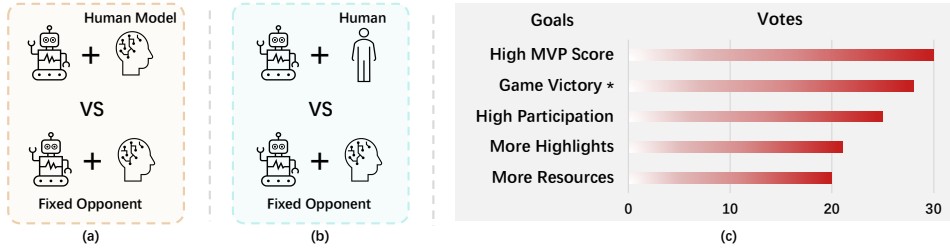

Figure 4: **Environment Setup. (a)** Simulated environment: the human model-agent game tests. **(b)** Real-world environment: the human-agent game tests. **(c)** Top 5 goals based on the stats of our participant survey. * denotes the task goal. The participant survey contains 8 initial goals, each participant can vote up to 5 non-repeating goals, and can also add additional goals. 30 participants voluntarily participated in the voting.

---

[1] All experiments are conducted subject to oversight by an Institutional Review Board (IRB).

**Training Setup:** We were authorized by Ye et al. to use the Wukong agent (Ye et al., 2020a) as the pre-trained agent and use the JueWu-SL agent (Ye et al., 2020b) as the fixed human model. Note that both the Wukong agent and the JueWu-SL agent were developed at the same level as the high-level (top 1%) players. We adopted the top 4 human goals as $\mathcal{G}^H$ for the pre-trained agent to enhance the human model. The corresponding goal reward function can be found in Appendix B.3. We trained the human primitive value network and fine-tune the agent until they converge for 12 and 40 hours, respectively, using a physical computer cluster with 49600 CPU cores and 288 NVIDIA V100 GPU cards. The batch size of each GPU is set to 256. The hyper-parameters $\alpha$ and $\beta$ are set to 2 and 50, respectively. The step size and unit size of the LSTM module are set to 16 and 4096, respectively.

**Baseline Setup:** We compared the RLHG agent with two baseline agents: the Wukong agent (Ye et al., 2020a) and the Human Reward Enhancement (HRE) agent.

- **Wukong:** A state-of-the-art agent in *HoK*, trained using the PPO algorithm to optimize the Game Victory goal in an agent-only team setting.
- **HRE:** An agent that is fine-tuned from the pre-trained Wukong agent by directly incorporating the human-centered objective into the optimization objective (see Eq. 3).
- **RLHG:** An agent that is fine-tuned from the pre-trained Wukong agent by incorporating the human-centered objective based on positive human gains into the optimization objective.

The human model-agent team (4 Wukong agents plus 1 human model) was adopted as the fixed opponent for all tests. For fair comparisons, both the HRE and RLHG agents were trained using the same human reward function, and all common parameters, network structures, and training resources were kept consistent. Results were reported over five random seeds.

## 4.2 HUMAN MODEL-AGENT TEST

Directly evaluating agents with humans is expensive, which is not conducive to model selection and iteration. Instead, we build a simulated environment, i.e., human model-agent game tests, to evaluate agents, in which the human model, i.e., the JueWu-SL agent, teams up with different agents. The results of the human model on different goal after teaming up with different agents are shown in Figure 5, including the top 4 human goals and the task goal.

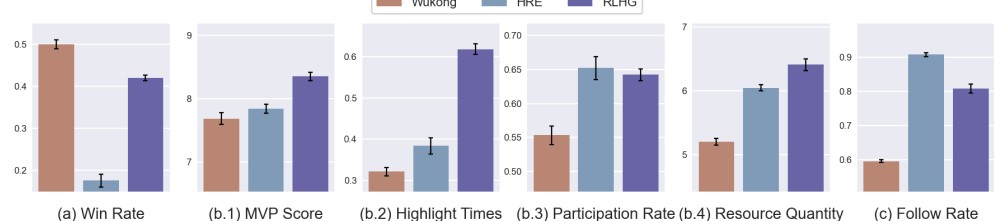

Figure 5: The performance of the human model in achieving human goals after teaming up with different agents. **(a)** The task goal. **(b)** The top 4 human goals (b.1, b.2, b.3, and b.4). **(c)** The follow rate metric: the frequency with which an agent follows a human in the entire game. Each agent played 10,000 games. Error bars represent 95% confidence intervals, calculated over games.

From Figure 5 (b), we can observe that both the RLHG agent and the HRE agent significantly enhance the performance of the human model in achieving the top 4 human goals, and the RLHG agent has achieved the best enhancement effect on most of the human goals. These results validate that the human-centered objective can encourage agents to learn behaviors that better enhance humans in achieving their goals.

However, as shown in Figure 5 (a), the HRE agent drops significantly on the task goal. We observed the actual performance of the HRE agent and found that the HRE agent did many unreasonable behaviors. For example, to assist the human model in achieving the Participation Rate and Highlight Times goals, the HRE agent had been following the human model throughout the entire game. Such excessive following behaviors greatly affected its original abilities to complete the task and led to a decreased Win Rate. This can also be reflected in Figure 5(c), in which the HRE agent has the highest Follow Rate metric. Although the Follow Rate of the RLHG agent has also increased, we observed that most of the following behaviors of the RLHG agent can effectively assist the human model. We also observed that the Win Rate of the RLHG agent slightly decreased, which is consistent

with expectations, as enhancing the abilities to achieve human goals inevitably sacrifices the abilities to achieve the task goal. We implement an adaptive adjustment mechanism to balance the two optimization objectives. We simply utilize the agent's original value network to measure the degree of completing the task goal and set the task gate to 1 (enhancing the human) when the original value is above the specified threshold $\xi$, and to 0 (completing the task) otherwise. The threshold $\xi$ depends on the human preference, i.e. the relative importance of the task goal and the human goals. We verify the effectiveness of the adaptive adjustment mechanism in Appendix C.3.

## 4.3 HUMAN-AGENT TEST

In this section, we conduct online experiments to examine whether the RLHG agent can effectively enhance the experience of human participants. Note that, We did not compare the HRE agent, since the Win Rate of the HRE agent is extremely low. We used a within-participant design for the experiment: each participant teams up with four agents. This design allowed us to evaluate both objective performance as well as subjective preference. All participants read detailed guidelines and provided informed consent before the testing. Each participant tested 20 matches. After each test, participants reported their preference over their agent teammates. For fair comparisons, participants were not told the type of their agent teammates.

Table 1: The results of **high-level** participants achieving goals after teaming up with different agents. Results for the task goal are expressed as percentages, and results for human goals are expressed as mean (std.).

| Agent \ Goals | Task Goal | Top 4 human Goals | | |
| --- | --- | --- | --- | --- |
| | Win Rate | MVP Score | Highlight Times | Participation Rate | Resource Quantity |
| Wukong | 52% | 8.86 (0.79) | 0.53 (0.21) | 0.46 (0.11) | 5.3 (2.87) |
| RLHG | 46.7% | **10.28** (0.75) | **0.87** (0.29) | **0.58** (0.09) | **6.28** (2.71) |

Table 2: The results of **general-level** participants achieving goals after teaming up with different agents. Results for the task goal are expressed as percentages, and results for human goals are expressed as mean (std.).

| Agent \ Goals | Task Goal | Top 4 Human Goals | | |
| --- | --- | --- | --- | --- |
| | Win Rate | MVP Score | Highlight Times | Participation Rate | Resource Quantity |
| Wukong | 34% | 7.44 (0.71) | 0.37 (0.349) | 0.41 (0.11) | 4.98 (2.73) |
| RLHG | 30% | **9.1** (0.61) | **0.75** (0.253) | **0.59** (0.05) | **5.8** (2.78) |

We first compare the objective performance of the participants on different human goal metrics after teaming up with different agents. Table 1 and Table 2 demonstrate the results of high-level and general-level participants, respectively. We see that both high-level and general-level participants had significantly improved their performance on all top 4 human goals after teaming up with the RLHG agent. Notably, the RLHG agent effectively improved the performance of general-level participants in achieving human goals even better than the primitive performance of high-level participants. We also notice that the Win Rate of the participants decreased when they teamed up with the RLHG agent, which is consistent with the results in the simulated environment. However, we find in the subsequent subjective preference statistics that the improvement of Gaming Experience brought by the enhancement outweighs the negative impact of the decrease in Win Rate.

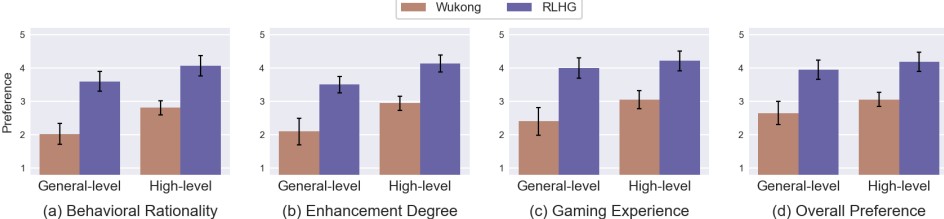

Figure 6: **Participants' preference over their agent teammates.** (a) Behavioral Rationality: the reasonableness of the agent's behavior. (b) Enhancement Degree: the degree to which the agent enhances your abilities to achieve goals. (c) Gaming Experience: your overall gaming experience. (d) Overall Preference: your overall preference for your agent teammates. Participants scored (1: Terrible, 2: Poor, 3: Normal, 4: Good, 5: Perfect) in these metrics after each game test. Error bars represent 95% confidence intervals, calculated over games.

We then compare the subjective preference metrics, i.e., the Behavioral Rationality, the Enhancement Degree, the Gaming Experience, and the Overall Preference, reported by participants over their agent teammates, as shown in Figure 6. We find that most participants showed great interest in the RLHG agent, and they believed that the RLHG agent's enhancement behaviors were more reasonable than that of the Wukong agent, and the RLHG agent's enhancement behaviors brought them a better gaming experience. A high-level participant commented on the RLHG agent "The agent frequently helps me do what I want to do, and this feeling is amazing." In general, participants were satisfied with the RLHG agent and gave higher scores in the Overall Preference metric (Figure 6(d)).

### 4.4 CASE STUDY

To better understand how the RLHG agent effectively enhances the experience of participants, we visualize the values predicted by the gain network in two test scenarios where participants benefitted from the RLHG agent's assistance, as illustrated in Figure 7. In the first scenario (Figure 7(a)), the RLHG agent successfully assisted the participant in achieving the highlight goal, whereas the Wukong agent disregarded the participant, leading to a failure in achieving the highlight goal. Figure 7(b) illustrates that the gain of the RLHG agent, when accompanying the participant, is positive, reaching the maximum when the participant achieved the highlight goal. In the second scenario (Figure 7(c)), the RLHG agent actively relinquishes the acquisition of the monster resource, enabling the participant to successfully achieve the resource goal. Conversely, the Wukong agent competes with the participant for the monster resource, resulting in the participant's failure to achieve the resource goal. Figure 7(d) also reveals that the gain of the RLHG agent's behavior - actively forgoing the monster resource, is positive, with the gain peaking when the participant achieved the resource goal. These results indicate that the RLHG agent learns effective enhancement behaviors through the guidance of the gain network.

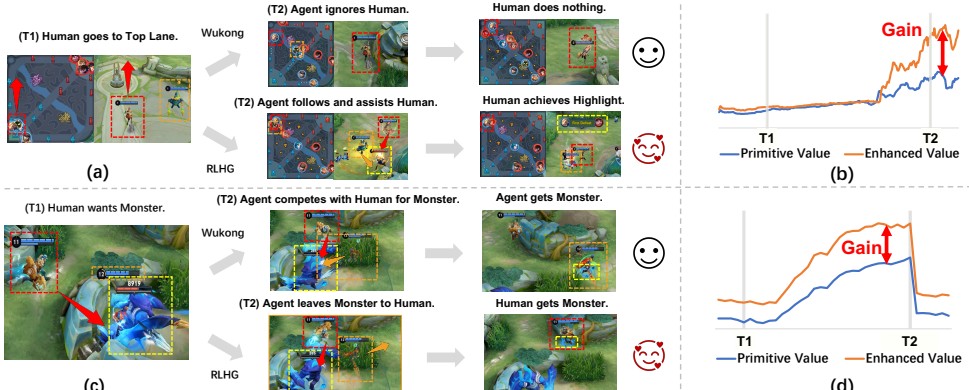

Figure 7: **The RLHG agent enhances the experience of participants in two scenarios. (a)** The Wukong agent ignores the participant; The RLHG agent accompanies the participant and assists the participant in achieving the highlight goal. **(b)** The gain value in scenario (a). **(c)** The Wukong agent competes with the participant for the monster resource; The RLHG agent actively forgoes the monster resource, and the participant successfully achieves the resource goal. **(d)** The gain value in scenario (c).

## 5 CONCLUSION

In this work, we proposed a "human-centered" modeling scheme for collaborative agents, designed to enhance the human experience. We represent the human experience as the goals they expect to achieve during the task. To enhance the extent to which humans achieve these goals while maintaining agent's original abilities, we introduced the RLHG approach. The RLHG approach initially trains a value network to estimate the primitive expected human return in achieving human goals, utilizing episodes collected by directly partnering the human and the self-centered agent for execution. Subsequently, the RLHG approach trains a gain network to estimate the expected positive gain of human return when subjected to effective enhancement, compared to the "baseline." The agent is fine-tuned using a combination of its original advantage and the human-centered advantage calculated by the positive human gains. We conducted real-world human-agent tests in *Honor of Kings*, and the results in both objective performance and subjective preference demonstrate that the RLHG agent offers a superior gaming experience for humans.

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

# A  ENVIRONMENT DETAILS

## A.1  GAME INTRODUCTION

MOBA (Multiplayer Online Battle Arena) games (Silva & Chaimowicz, 2017), characterized by multi-agent cooperation and competition mechanisms, long time horizons, enormous state-action spaces ($10^{20000}$, (Wu, 2019)), and imperfect information (OpenAI et al., 2019; Ye et al., 2020a), have attracted much attention from researchers. The general layout of the MOBA games is depicted in the Figure 8(a). The red and blue on the map indicate two teams, and the map can be divided into 3 *lanes* (i.e., yellow areas separated into the top, middle, and bottom lane), 4 *jungle areas* (i.e. green areas), and 2 *base areas*. Circles symbolize *turrets*.

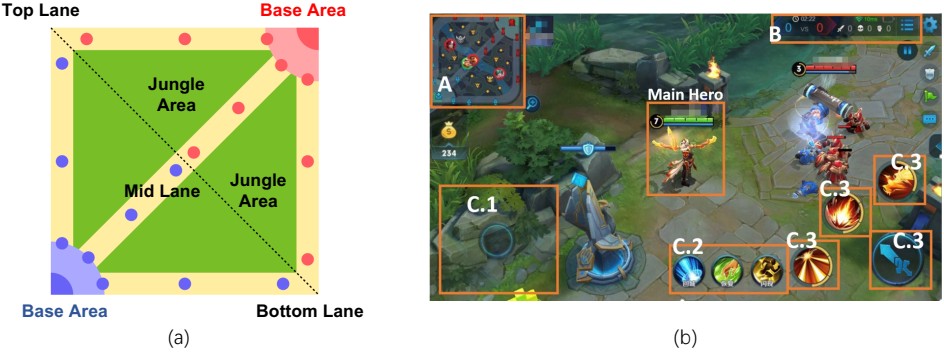

Figure 8: **(a)** A generic map of MOBA games. **(b)** The UI interface of *Honor of Kings*.

*Honor of Kings* (Wei et al., 2022) is a typical MOBA game often used as a testbed for game AI research (Wu, 2019; Ye et al., 2020a;b;c; Gao et al., 2021; 2022). The game is played by two opposing teams on a symmetrical map, each comprising five players. The game environment depicted in Figure 8(b) comprises the main hero with peculiar skill mechanisms and attributes, controlled by each player. The player can maneuver the hero's movement using the bottom-left wheel (C.1) and release the hero's skills through the bottom-right buttons (C.2, C.3). The player can view the local environment on the screen, the global environment on the top-left mini-map (A), and access game stats on the top-right dashboard (B). Players of each camp compete for resources through team confrontation and collaboration, etc., with the task goal of winning the game by destroying the crystal in the opposing team's base area.

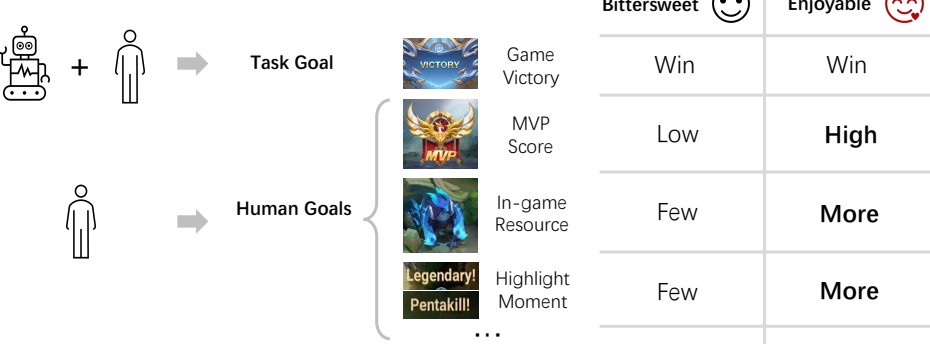

Figure 9: In-game goals, based on our participant survey (see Figure 24 (c)). Human players pursue multiple goals for more enjoyable gaming experience.

The gaming experience is crucial for a player's engagement and satisfaction. Along with the task goal, players also pursue multiple individual goals (Figure 9, such as achieving a higher MVP score, experiencing more highlight moments, and obtaining more in-game resources. The pursuit of these

goals can contribute to a more enjoyable and rewarding gaming experience. Agents can enhance the gaming experience of their human partners through interactive behaviors such as timely support, sharing resources, and active protection.

For fair comparisons, all experiments in this paper were carried out using a fixed released game engine version (Version 8.2 series) of *Honor of Kings*.

## A.2    HERO POOL

Table 3 shows the full hero pool used in Experiments. Each match involves two camps playing against each other, and each camp consists of five randomly picked heroes.

Table 3: Hero pool used in **Experiments**.

| | |
|---|---|
| Full Hero pool | Lian Po, Xiao Qiao, Zhao Yun, Mo Zi, Da Ji, Ying Zheng, Sun Shangxiang, Luban Qihao, Zhuang Zhou, Liu Chan Gao Jianli, A Ke, Zhong Wuyan, Sun Bin, Bian Que, Bai Qi, Mi Yue, Lv Bu, Zhou Yu, Yuan Ge, Chengji Sihan Xia Houdun, Zhen Ji, Cao Cao, Dian Wei, Gongben Wucang, Li Bai, Make Boluo, Di Renjie, Da Mo, Xiang Yu Wu Zetian, Si Mayi, Lao Fuzi, Guan Yu, Diao Chan, An Qila, Cheng Yaojin, Lu Na, Jiang Ziya, Liu Bang, Chang E Han Xin, Wang Zhaojun, Lan Lingwang, Hua Mulan, Ai Lin, Zhang Liang, Buzhi Huowu, Nake Lulu, Ju Youjing Ya Se, Sun Wukong, Niu Mo, Hou Yi, Liu Bei, Zhang Fei, Li Yuanfang, Yu Ji, Zhong Kui, Yang Yuhuan, Zhu Bajie Yang Jian, Nv Wa, Ne Zha, Ganjiang Moye, Ya Dianna, Cai Wenji, Taiyi Zhenren, Donghuang Taiyi, Gui Guzi Zhu Geliang, Da Qiao, Huang Zhong, Kai, Su Lie, Baili Xuance, Baili Shouyue, Yi Xing, Meng Qi, Gong Sunli Shen Mengxi, Ming Shiyin, Pei Qinhu, Kuang Tie, Mi Laidi, Yao, Yun Zhongjun, Li Xin, Jia Luo, Dun Shan, Sun Ce Shangguan Waner, Ma Chao, Dong Fangyao, Xi Shi, Meng Ya, Luban Dashi, Pan Gu, Meng Tian, Jing, A Guduo Xia Luote, Lan, Sikong Zhen, Erin, Yun ying, Jin Chan, Fei, Sang Qi, Ge Ya, Hai Yue, Zhao Huaizhen, Lai Xiao |

## B    FRAMEWORK DETAILS

### B.1    INFRASTRUCTURE DESIGN

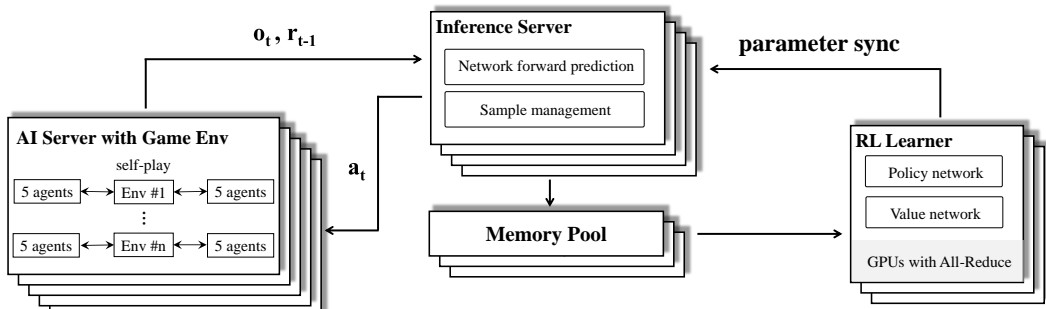

Figure 10: The infrastructure of the training system.

Figure 10 shows the infrastructure of the training system (Ye et al., 2020a), which consists of four key components: AI Server, Inference Server, RL Learner, and Memory Pool. The AI Server (the actor) covers the interaction logic between the agents and the environment. The Inference Server is used for the centralized batch inference on the GPU side. The RL Learner (the learner) is a distributed training environment for RL models. And the Memory Pool is used for storing the experience, implemented as a memory-efficient circular queue.

## B.2 TASK REWARD DESIGN

Table 4 demonstrates the details of the designed task reward from environment.

Table 4: The details of the environment reward.

| Head | Reward Item | Weight | Type | Description |
|---|---|---|---|---|
| Farming Related | Gold | 0.005 | Dense | The gold gained. |
| | Experience | 0.001 | Dense | The experience gained. |
| | Mana | 0.05 | Dense | The rate of mana (to the fourth power). |
| | No-op | -0.00001 | Dense | Stop and do nothing. |
| | Attack monster | 0.1 | Sparse | Attack monster. |
| KDA Related | Kill | 1 | Sparse | Kill a enemy hero. |
| | Death | -1 | Sparse | Being killed. |
| | Assist | 1 | Sparse | Assists. |
| | Tyrant buff | 1 | Sparse | Get buff of killing tyrant, dark tyrant, storm tyrant. |
| | Overlord buff | 1.5 | Sparse | Get buff of killing the overlord. |
| | Expose invisible enemy | 0.3 | Sparse | Get visions of enemy heroes. |
| | Last hit | 0.2 | Sparse | Last hitting an enemy minion. |
| Damage Related | Health point | 3 | Dense | The health point of the hero (to the fourth power). |
| | Hurt to hero | 0.3 | Sparse | Attack enemy heroes. |
| Pushing Related | Attack turrets | 1 | Sparse | Attack turrets. |
| | Attack crystal | 1 | Sparse | Attack enemy home base. |
| Win/Lose Related | Destroy home base | 4 | Sparse | Destroy enemy home base. |

## B.3 HUMAN REWARD DESIGN

Table 5 demonstrates the details of the designed human reward.

Table 5: The details of the human reward.

| Head | Reward Item | Weight | Type | Description |
|---|---|---|---|---|
| MVP Score Related | Kill | 1 | Sparse | Kill a enemy hero. |
| | Death | -1 | Sparse | Being killed. |
| | Assist | 1 | Sparse | Assists. |
| | Hurt to hero | 0.3 | Sparse | Attack enemy heroes. |
| | Health point | 3 | Dense | The health point of the hero (to the fourth power). |
| Participation Related | Participation | 1 | Dense | Percentage of players participating in the team fight. |
| Highlight Related | Highlight | 2 | Sparse | Double kill, triple kill, quadra kill, penta kill. |
| Resource Related | Buff | 1 | Sparse | Get a red buff, blue buff. |
| | Health cake | 1 | Sparse | Get a health cake. |

## B.4 Feature Design

Table 6 shows the designed features of the Wukong agent (Ye et al., 2020a), some of which (observable) are used as human features.

Table 6: The observation space of agents. ∗ are used as human features.

| Feature Class | Field | Description | Dimension | Type |
|---|---|---|---|---|
| **1. Unit feature** | Scalar | Includes heroes, minions, monsters, and turrets | 8599 | |
| Heroes* | Status | Current HP, mana, speed, level, gold, KDA, buff, bad states, orientation, visibility, etc. | 1842 | (one-hot, normalized float) |
| | Position | Current 2D coordinates | 20 | (normalized float) |
| | Attribute | Is main hero or not, hero ID, camp (team), job, physical attack and defense, magical attack and defense, etc. | 1330 | (one-hot, normalized float) |
| | Skills | Skill 1 to Skill N's cool down time, usability, level, range, buff effects, bad effects, etc. | 2095 | (one-hot, normalized float) |
| | Item | Current item lists | 60 | (one-hot) |
| Minions | Status | Current HP, speed, visibility, killing income, etc. | 1160 | (one-hot, normalized float) |
| | Position | Current 2D coordinates | 80 | (normalized float) |
| | Attribute | Camp (team) | 80 | (one-hot) |
| | Type | Type of minions (melee creep, ranged creep, siege creep, super creep, etc.) | 200 | (one-hot) |
| Monsters* | Status | Current HP, speed, visibility, killing income, etc. | 868 | (one-hot, normalized float) |
| | Position | Current 2D coordinates | 56 | (normalized float) |
| | Type | Type of monsters (normal, blue, red, tyrant, overlord, etc.) | 168 | (one-hot) |
| Turrets | Status | Current HP, locked targets, attack speed, etc. | 520 | (one-hot, normalized float) |
| | Position | Current 2D coordinates | 40 | (normalized float) |
| | Type | Type of turrets (tower, high tower, crystal, etc.) | 80 | (one-hot) |
| **2. In-game stats feature** | Scalar | Real-time statistics of the game | 68 | |
| Static statistics* | Time | Current game time | 5 | (one-hot) |
| | Gold | Golds of two camps | 12 | (normalized float) |
| | Alive heroes | Number of alive heroes of two camps | 10 | (one-hot) |
| | Kill | Kill number of each camp (Segment representation) | 6 | (one-hot) |
| | Alive turrets | Number of alive turrets of two camps | 8 | (one-hot) |
| Comparative statistics* | Gold diff | Gold difference between two camps (Segment representation) | 5 | (one-hot) |
| | Alive heroes diff | Alive heroes difference between two camps | 11 | (one-hot) |
| | Kill diff | Kill difference between two camps | 5 | (one-hot) |
| | Alive turrets diff | Alive turrets difference between two camps | 6 | (one-hot) |
| **3. Invisible opponent information** | Scalar | Invisible information used for the value net | 560 | |
| Opponent heroes | Position | Current 2D coordinates, distances, etc. | 120 | (normalized float) |
| NPC | Position | Current 2D coordinates of all non-player characters, including minions, monsters, and turrets | 440 | (normalized float) |
| **4. Spatial feature** | Spatial | 2D image-like, extracted in channels for convolution | 7x17x17 | |
| Skills* | Region | Potential damage regions of ally and enemy skills | 2x17x17 | |
| | Bullet* | Bullets of ally and enemy skills | 2x17x17 | |
| Obstacles* | Region | Forbidden region for heroes to move | 1x17x17 | |
| Bushes* | Region | Bush region for heroes to hide | 1x17x17 | |
| Health cake* | Region | Cake for heroes to recover blood | 1x17x17 | |

## B.5 ACTION DESIGN

Table 7 shows the action space of agents.

Table 7: The action space of agents.

| Action | Detail | Description |
|---|---|---|
| What | Illegal action | Placeholder. |
| | None action | Executing nothing or stopping continuous action. |
| | Move | Moving to a certain direction determined by move x and move y. |
| | Normal Attack | Executing normal attack to an enemy unit. |
| | Skill1 | Executing the first skill. |
| | Skill2 | Executing the second skill. |
| | Skill3 | Executing the third skill. |
| | Skill4 | Executing the fourth skill (only a few heroes have Skill4). |
| | Summoner ability | An additional skill choosing before the game begins (10 to choose). |
| | Return home(Recall) | Returning to spring, should be continuously executed. |
| | Item skill | Some items can enable an additional skill to player's hero. |
| | Restore | Blood recovering continuously in 10s, can be disturbed. |
| | Collaborative skill | Skill given by special ally heroes. |
| How | Move X | The x-axis offset of moving direction. |
| | Move Y | The y-axis offset of moving direction. |
| | Skill X | The x-axis offset of a skill. |
| | Skill Y | The y-axis offset of a skill. |
| Who | Target unit | The game unit(s) chosen to attack. |

## B.6 NETWORK ARCHITECTURE

Figure 11 shows the detailed network architecture of the RLHG agent, which consists of two parts: the pre-trained Wukong model (Ye et al., 2020a), and the human enhancement module.

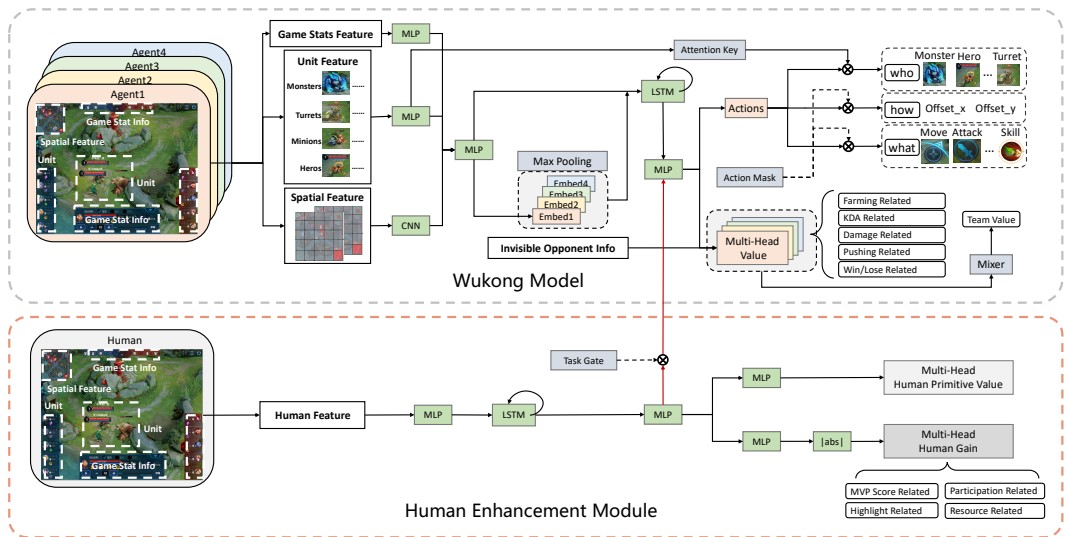

Figure 11: The network structure.

**Human Enhancement Module.** Human features are sequentially fed into the Fully-Connected (FC) layers with LSTM (Hochreiter & Schmidhuber, 1997) to extract human policy embedding. The policy embedding is used to predict human primitive values and gains. We apply the absolute activation function to ensure that the gains are non-negative. To manage the uncertain value of state-action in the game, we introduce the multi-head value estimation (Ye et al., 2020a) into the network by grouping the human reward in Table 5.

**Human Conditioned Policy Modeling.** We use surgery techniques (Chen et al., 2015; OpenAI et al., 2019) to fuse the human policy embedding into the agent's original network, i.e. adding more randomly initialized units to an internal FC layer. The task gate is used to control the agent's policy

style, i.e., for the non-enhancement mode, the task gate is set to 0, and for the enhancement mode, the task gate is set to 1. The agent's policy network predicts a sequence of actions for each agent based on its observation and human policy embedding.

**Network Parameter Details.** All hyper-parameters of the Wukong model are consistent with the original (Ye et al., 2020a). The unit size and step size of the LSTM module in the human enhancement module are set to 4096 and 16, respectively. The parameters of each FC layer are shown in our code. We use Adam (Kingma & Ba, 2014) with an initial learning rate of 0.0001 for fine-turning.

## B.7 ALGORITHM DETAILS

The more detailed pseudocode of RLHG is shown in Algorithm 2. The RLHG approach aims to fine-tune a pre-trained agent $\pi_0$ to enhance a given human model $\pi_H$. The RLHG algorithm consists of two stages: the Human Primitive Value Estimation stage and the Human Enhancement Training stage.

**Human Primitive Value Estimation:** The RLHG algorithm starts with initializing a human primitive value network $V_\phi$. Trajectories are collected by pairing the pre-trained agent $\pi_0$ and the human $\pi_H$ to perform in a collaborative environment. Next, these trajectories are utilized to compute the human return $G^H$ for achieving human goals $\mathcal{G}^H$. Lastly, the human primitive value network $V_\phi$ is updated using the human return by minimizing the Temporal Difference (TD) error. After $V_\phi$ converges, it is frozen and used as the baseline for calculating human gains in the next stage.

**Human Enhancement Training:** The RLHG algorithm initializes the agent's policy network $\pi_\theta$ and value network $V_\psi$ conditioned on the human policy $\pi_H$, and a gain network $\Delta_\omega$. Trajectories are then collected by pairing the agent $\pi_\theta$ and the human $\pi_H$ to perform in a collaborative environment. Subsequently, these trajectories are employed to compute the original return $G$ and human return $G^H$. The RLHG algorithm calculates the human gain from the human return based on the predicted human primitive value $V_\phi(s)$. This human gain is used for human-centered advantage calculations. The agent's policy network $\pi_\theta$ is fine-tuned using the Policy Gradient algorithm (Sutton et al., 1999; Mnih et al., 2016), like PPO (Schulman et al., 2017), which combines the original advantage $A$ and the human-centered advantage $\widehat{A}_H$. The agent's value network $V_\psi$ is fine-tuned using the agent's original return. Finally, the gain network is updated when the human gain is positive, as per Equation 5.

---

**Algorithm 2** Reinforcement Learning from Human Gain (RLHG)

---

**Require**: Human policy network $\pi_H$, human goals $\mathcal{G}^H$, agent policy network $\pi_0$, agent value network $V$, hyper-parameter $\alpha$

**Process**:

1: Initialize human primitive value network $V_\phi$;
   // Stage I: Human Primitive Value Estimation
2: **while** not converged **do**
3:     Freeze $\pi_0$ and collect human-agent team $< \pi_0, \pi_H >$ trajectories;
4:     Compute human return $G^H$ for achieving goals $\mathcal{G}^H$;
5:     Update human primitive value network $V_\phi \leftarrow G^H$
6: **end while**
7: Initialize agent policy network $\pi_\theta \leftarrow \pi_0$, agent value network $V_\psi \leftarrow V$, human gain network $\Delta_\omega$;
   // Stage II: Human Enhancement Training
8: **while** not converged **do**
9:     Freeze $V_\phi$ and collect human-agent team $< \pi_\theta, \pi_H >$ trajectories;
10:    Compute agent original return $G$ and human return $G_H$;
11:    Compute agent self-centered advantage $A = G - V_\psi$;
12:    Compute human gain $\Delta = G^H - V_\phi$
13:    Compute human-centered advantage $\widehat{A}_H = \Delta - \Delta_\omega$;
14:    Update agent policy network $\pi_\theta \leftarrow A + \alpha \cdot \widehat{A}_H$;
15:    Update agent value network $V_\psi \leftarrow G$;
16:    **if** $\Delta > 0$ **then**
17:        Update human gain network $\Delta_\omega \leftarrow \Delta$;
18:    **end if**
19: **end while**

---

## C   SUPPLEMENTARY EXPERIMENT

### C.1   BASELINE DETAILS

We describe the training process of two baseline agents here, including the Wukong agent and the Human Reward Enhancement (HRE) agent.

**Wukong** (Ye et al., 2020a): A state-of-the-art agent in *Honor of Kings*, which can easily beat the high-level human players. As shown in Figure 12, the agent is trained in agent-only team settings, with the optimization objective being to maximize Game Victories.

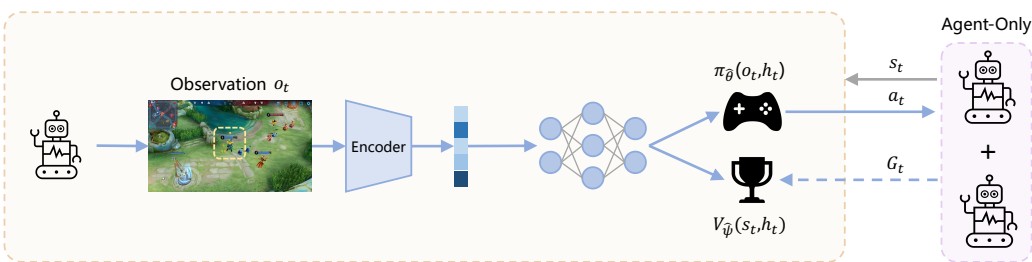

Figure 12: **The Wukong training process.** The policy network $\pi_{\hat{\theta}}$ and value network $V_{\hat{\psi}}$ are trained in the agent-only team settings.

To avoid instability in training within large-scale distributed environments, Wukong uses the Dual-clip PPO algorithm to train the policy network $\pi_{\hat{\theta}}$, which is a practical improvement of the PPO algorithm (Schulman et al., 2017). When $\pi_{\hat{\theta}}(a|s) \gg \pi_{\hat{\theta}_{old}}(a|s)$ and $A < 0$, the radio $r(\hat{\theta}) = \frac{\pi_{\hat{\theta}}(a|s)}{\pi_{\hat{\theta}_{old}}(a|s)}$ is huge, which causes the large and unbounded variance since $r(\hat{\theta}) \cdot A \ll 0$. Dual-clip PPO introduces another clipping parameter $c$ that indicates the lower bound when $A < 0$. The policy loss is the following:

$$L^{\pi}(\hat{\theta}) = \mathbb{E}_{s \in S, a \in A}\left[\max(cA, \min(\text{clip}(r(\hat{\theta}), 1 - \epsilon, 1 + \epsilon)A, r(\hat{\theta})A)\right],$$

where $\epsilon$ is the original clip parameter in PPO. In the experiments, we use the same parameters as Wukong, with the two clipping hyperparameters $\epsilon$ and $c$ set to 0.2 and 3, respectively. The discount factor $\gamma$ is set as 0.998. Besides, Wukong uses Generalized Advantage Estimation (GAE) (Schulman et al., 2015) for return calculation, with $\lambda = 0.95$ to reduce the variance caused by delayed effects.

To decrease the variance of value estimation, Wukong uses full information about the game state, including observations hidden from the policy, as input to the value network $V_{\hat{\psi}}$ in training. To estimate the value of the ever-changing game state more accurately, Wukong introduces multi-head value by decomposing the reward. The multi-head value loss is:

$$L^V(\hat{\theta}) = \mathbb{E}_{s \in S}\left[\sum_k (G^k - V_{\hat{\theta}}^k(s))\right],$$

where $G^k$ and $V_{\hat{\theta}}^k$ are the discounted return and value estimation of the $k$-th head, respectively. Then, the total value estimation is the weighted sum of the head value estimates $V_{total}(s) = \sum_k w_k \cdot V_{\hat{\theta}}^k(s)$, where $w_k$ is the weight of the $k$-th head.

**HRE** (Human Reward Enhancement): An agent that is fine-tuned from the pre-trained Wukong agent by directly incorporating the human-centered objective into the optimization objective. The optimization objective can be formulated as:

$$J(\theta) = V^{\pi_\theta, \pi_H}(s) + \alpha \cdot V_H^{\pi_\theta, \pi_H}(s) = \mathbb{E}_{\pi_\theta, \pi_H}\left[G_t + \alpha \cdot G_t^H | s_t = s\right],$$

where $\alpha$ is a balancing parameter. The approximation to the policy gradient is defined as follows:

$$\nabla J(\theta) = \mathbb{E}_{\pi_\theta, \pi_H}\left[\sum_{t=0}^{\infty} \nabla_\theta \log \pi_\theta(a_t|o_t)\left(A(s_t, a_t) + \alpha \cdot A_H(s_t, a_t)\right)\right],$$

where $A(s, a) = \mathbb{E}_{\pi_\theta, \pi_H}[G_t | s_t = s, a_t = a] - V^{\pi_\theta, \pi_H}(s)$ is the self-centered advantage and $A_H(s, a) = \mathbb{E}_{\pi_\theta, \pi_H}[G_t^H | s_t = s, a_t = a] - V_H^{\pi_\theta, \pi_H}(s)$ is the human-centered advantage.

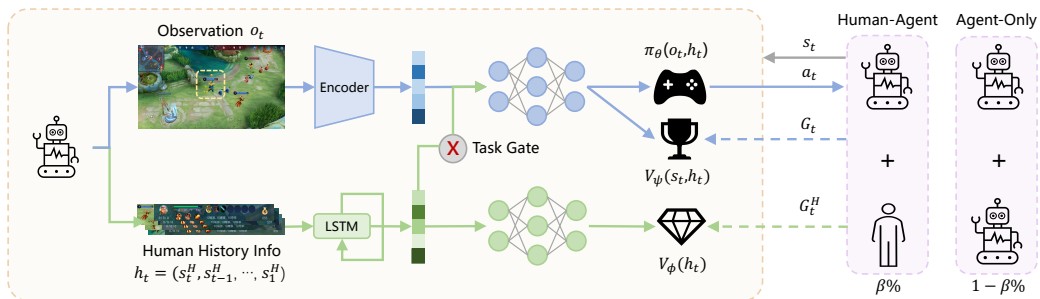

Figure 13: **The HRE training process.** The policy network $\pi_\theta$ and value network $V_\psi$ conditioned on the human policy $\pi_H$ are trained in $1 - \beta\%$ agent-only team settings and $\beta\%$ human-agent team settings. The human value network $V_\phi$ is trained to estimate the expected human return in human-agent team settings.

In comparison to Wukong (see Figure 13), HRE introduces a value network $V_\phi(s)$ to estimate the human value $V_H^{\pi_\theta, \pi_H}(s)$ when teaming up with agent policy $\pi_\theta$, which is utilized to calculate the human-centered advantage. The difference between HRE and RLHG is that HRE lacks modeling of human gains. Apart from these differences, other settings remain consistent with RLHG. Firstly, the policy network $\pi_\theta$ and value network $V_\psi$ are also conditioned on the human policy $\pi_H$. Secondly, the agent is trained in both $\beta\%$ human-agent team settings and $1 - \beta\%$ agent-only team settings. The human value network $V_\phi$ is only trained in human-agent team settings by minimizing the following loss function:

$$L(\phi) = \mathbb{E}_{s \in S, a \in A}\left[\|G^H - V_\phi(s)\|_2\right].$$

## C.2 Ablation Study

We examine the influence of the balance parameter $\alpha$, i.e., the relative importance of human goals relative to the task goal. The results of RLHG agents trained with different values of $\alpha$ are shown in Figure 14. We can see that with the increase of $\alpha$, the human model's performance in achieving human goals is significantly improved, but the negative effect is that the agent sacrifices its original ability to achieve the task goal (The Win Rate metric is reduced). We also notice that when $\alpha$ is too large, the Win Rate is significantly reduced, which will also have a negative impact on the MVP score goal. We find that when $\alpha$ is set to 2, it not only greatly improves the human model's performance in achieving human goals, but also has little impact on the Win Rate. Therefore, in our experiments, $\alpha$ is set to 2.

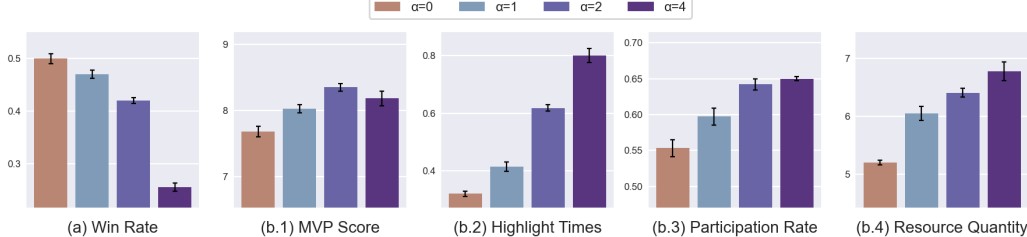

Figure 14: Influence of the balance parameter $\alpha$. Note that $\alpha = 0$ means training without enhancement.

## C.3 Adaptive Adjustment Mechanism

We implement an adaptive adjustment mechanism by simply utilizing the agent's original value network to measure the degree of completing the task goal. We first normalize the output of the original value network and then set the task gate to 1 (enhancing the human) when the normalized

value is above the specified threshold $\xi$, and to 0 (completing the task) otherwise. The threshold $\xi$ is used to control the timing of enhancement. The results of RLHG agents with different values of $\xi$ are shown in Figure 15. We can see that as the threshold $\xi$ increases, the Win Rate increases, and the human model's performance on human goals decreases. In practical applications, the threshold $\xi$ can be set according to human preference.

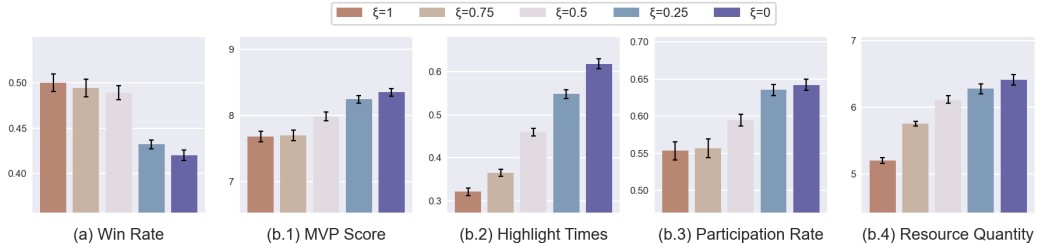

Figure 15: Influence of the threshold $\xi$. Note that $\xi = 1$ means never enhancement, and $\xi = 0$ means always enhancement.

## D  DETAILS OF HUMAN-AGENT COLLABORATION TEST

### D.1  ETHICAL REVIEW

The ethics committee of a third-party organization conducted an ethical review of our project. They reviewed our experimental procedures and risk avoidance methods (see Appendix D.1.3). They believed that our project complies with the "New Generation of AI Ethics Code" [2] of the country to which the participants belonged (China), so they approved our study. In addition, all participants consented to the experiment and provided informed consent (see Appendix D.1.1) for the study.

### D.1.1  INFORMED CONSENT

All participants were told the following experiment guidelines before testing:

- This experiment is to study human-agent collaboration technology in MOBA games.
- Your identity information will not be disclosed to anyone.
- All game statistics are only used for academic research.
- You will be invited into matches where your opponents and teammates are agents.
- Your goal is to win the game as much as possible by collaborating with agent teammates.
- Your agent teammates will assist you in achieving your individual goals in the game.
- After each test, you can report your gaming experience and express your preferences regarding the agent teammates.
- After each test, you may also voluntarily fill out a debriefing questionnaire to tell us your open-ended feedback about the agent teammates.
- Each game lasts 10-20 minutes.
- You may voluntarily choose whether to take the test. You can terminate the test at any time if you feel unwell during the test.
- At any time, if you want to delete your data, you can contact the game provider directly to delete it.

If participants volunteer to take the test, they will first provide written informed consent, then we will provide them with the equipment and game account, and explain the experimental details on the screen.

---

[2]China: MOST issues New Generation of AI Ethics Code, https://www.dataguidance.com/news/china-most-issues-new-generation-ai-ethics-code

### D.1.2 SCREENSHOTS

Screenshots of detailed experimental instructions are shown below.

1. Read tutorial and instruction on the study and gameplay. (Figure 16)
2. Read the detailed test content and precautions. (Figure 17)
3. Play the game with agents until the game is complete. (Figure 18)
4. Answer questions about perceptions and preferences.(Figure 19, 20, 21, and 22)
5. Volunteer to complete a debriefing questionnaire regarding open-ended feedback from your agent teammates. (Figure 23)
6. Repeat steps 3, 4, and 5 for a total of 20 times.

After the participant has read it carefully and confirmed complete understanding, the test will begin.

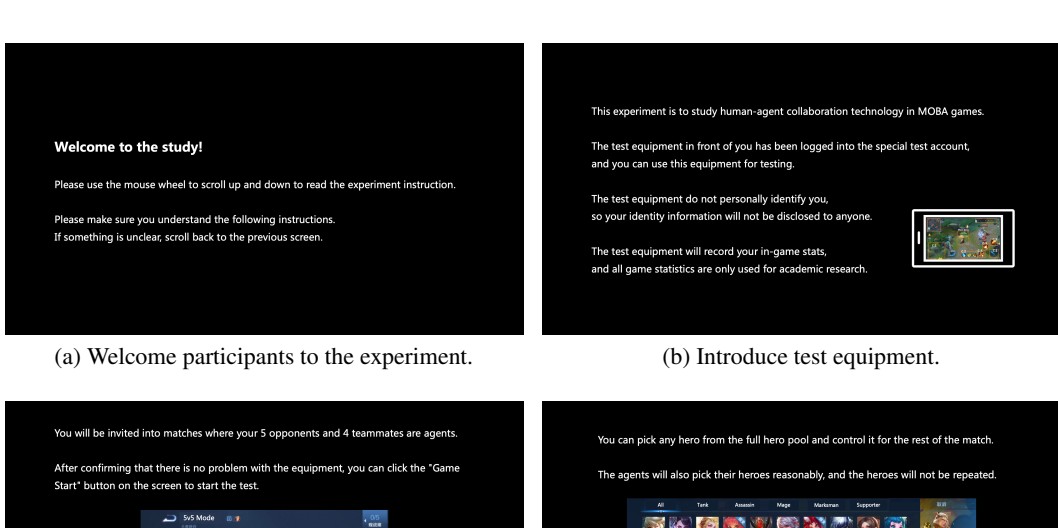

(a) Welcome participants to the experiment.                    (b) Introduce test equipment.

(c) Introduces test mode.                    (d) Introduce participant's controllable hero.

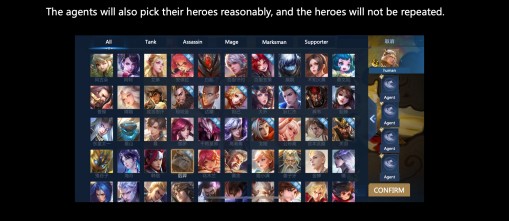

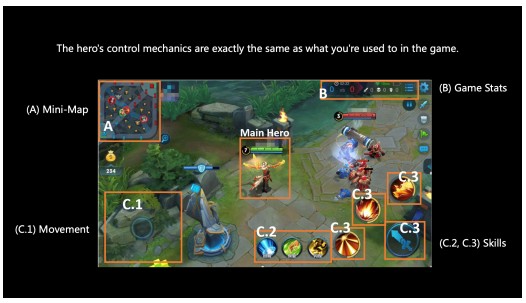

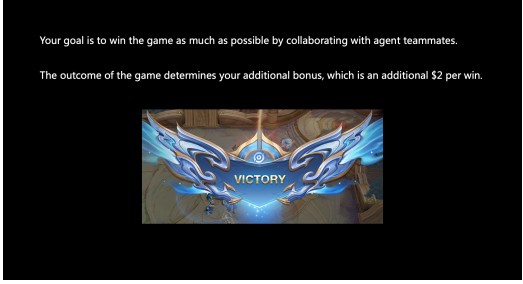

(e) Introduce the control mechanism.                    (f) Explain the task goal of the game.

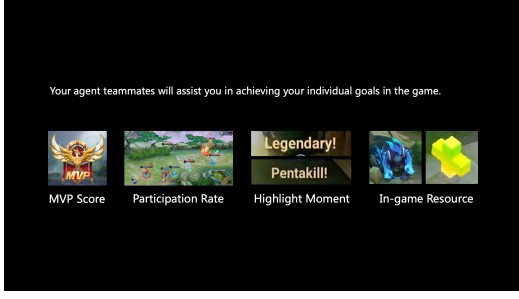

(g) Explain the enhanced individual goals.

Figure 16: Screenshots of tutorial and instruction screens.

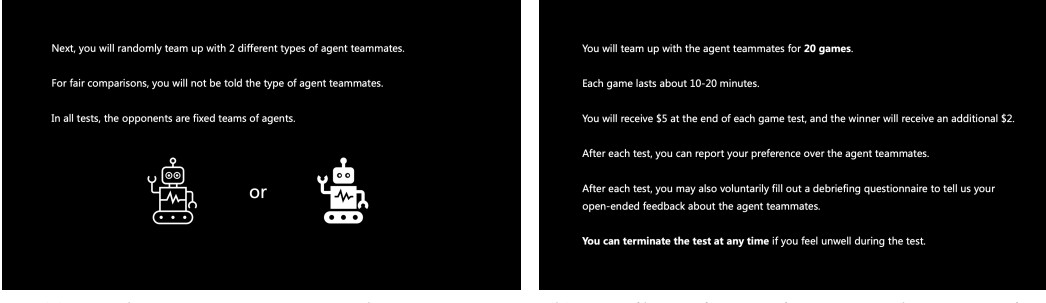

(a) Introduce agent teammates and opponents.  (b) Describe testing requirements and compensation.

Figure 17: Screenshot of experiment content.

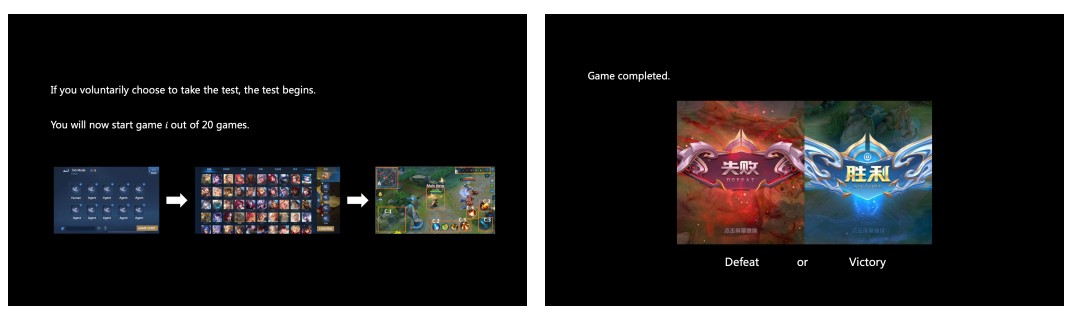

(a) Repeat the following process to test.  (b) Confirm completion of each test.

Figure 18: Screenshots of each test.

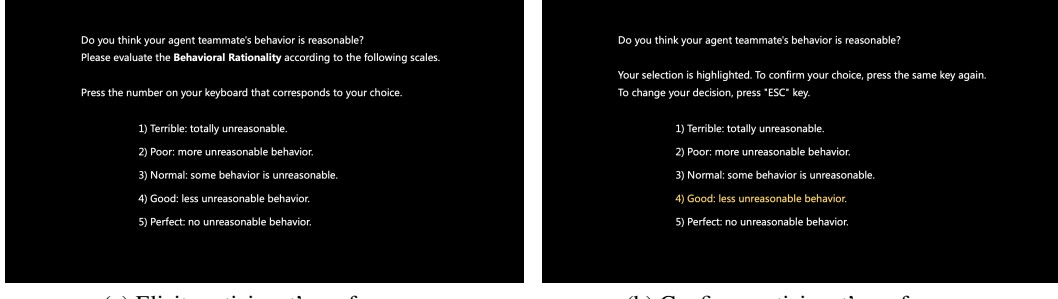

(a) Elicit participant's preference.  (b) Confirm participant's preference.

Figure 19: Screenshots of Behavioral Rationality elicitation over each test.

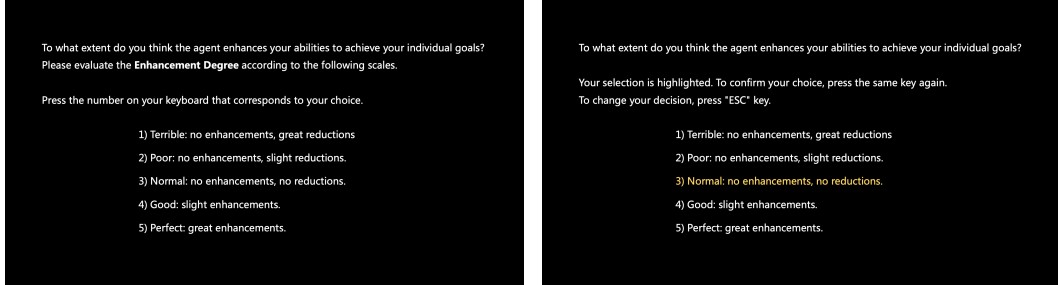

(a) Elicit participant's preference.      (b) Confirm participant's preference.

Figure 20: Screenshots of Enhancement Degree elicitation over each test.

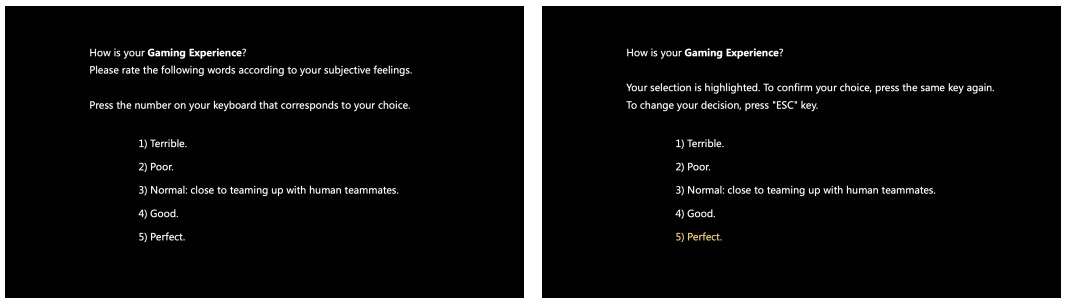

(a) Elicit participant's preference.      (b) Confirm participant's preference.

Figure 21: Screenshots of Gaming Experience elicitation over each test.

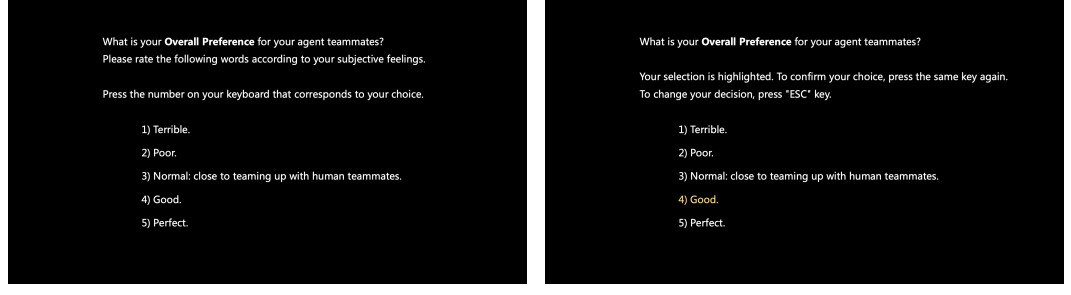

(a) Elicit participant's preference.      (b) Confirm participant's preference.

Figure 22: Screenshots of Overall Preference elicitation over each test.

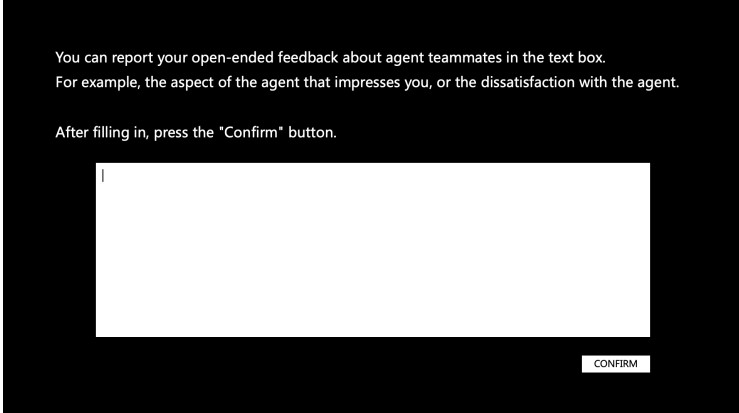

Figure 23: Screenshot of open-ended feedback about the agent teammates from debrief questionnaire.

### D.1.3 POTENTIAL PARTICIPANT RISKS

First, we analyze the risks of this experiment to the participants. The potential participant risks of the experiment mainly include the leakage of identity information and the time cost. And we have taken a series of measures to minimize these risks.

**Identity Information.** A series of measures have been taken to avoid this risk:

- All participants will be recruited with the help of a third party (the game provider of *Honor of Kings*), and we do not have access to participants' identities.
- We make a risk statement for participants and sign an identity information confidentiality agreement under the supervision of a third party.
- We only use unidentifiable game statistics in our research, which are obtained from third parties.
- Special equipment and game accounts are provided to the participants to prevent equipment and account information leakage.
- The identity information of all participants is not disclosed to the public.

**Time Cost.** We will pay participants to compensate for their time costs. Participants receive $5 at the end of each game test, and the winner will receive an additional $2. Each game test takes approximately 10 to 20 minutes, and participants can get about an average of $20 an hour.

### D.2 EXPERIMENTAL DETAILS

#### D.2.1 PARTICIPANT DETAILS

To conduct our experiments, we communicated with the game provider and obtained testing authorization. The game provider assisted in recruiting 30 experienced participants with anonymized personal information, which comprised 15 high-level (top 1%) and 15 general-level (top30%) participants. All participants have more than three years of experience in *Honor of Kings* and promise to be familiar with all mechanics in the game.

And special equipment and game accounts are provided to each participant to prevent equipment and account information leakage. The game statistics we collect are only for experimental purposes and are not disclosed to the public.

#### D.2.2 EXPERIMENTAL DESIGN

We used a within-participant design for the experiment: each participant teams up with four agents. This design allowed us to evaluate both objective performance as well as subjective preference. All participants read detailed guidelines and provided informed consent before the testing. Each participant tested 20 matches. Each participant is asked to randomly team up with two different

types of agents: the Wukong agent and the RLHG agent. After each test, participants reported their preference over their agent teammates. For fair comparisons, participants were not told the type of their agent teammates. The human model-agent team (4 Wukong agents plus 1 human model) was adopted as the fixed opponent for all tests.

In addition, as mentioned in Ye et al. (2020a); Gao et al. (2021), the response time of agents is usually set to 193ms, including observation delay (133ms) and response delay (60ms). The average APM of agents and top e-sport players are usually comparable (80.5 and 80.3, respectively). To make our test results more accurate, we adjusted the agents' capability to match the performance of high-level humans by increasing the observation delay (from 133ms to 200ms) and response delay (from 60ms to 120 ms).

### D.2.3 PARTICIPANT SURVEY DESCRIPTION

We designed an IRB-approved participant survey on what top 5 goals participants want to achieve in-game. The participant survey contains 8 initial goals, including Game Victory, High MVP Score, More Highlights, More Kill Counts, Few Death Counts, High Participation, More Resources, and More Visible Information. Each participant can vote up to 5 non-repeating goals, and can also add additional goals. 30 participants voluntarily participated in the voting, and the result is shown in Figure 24.

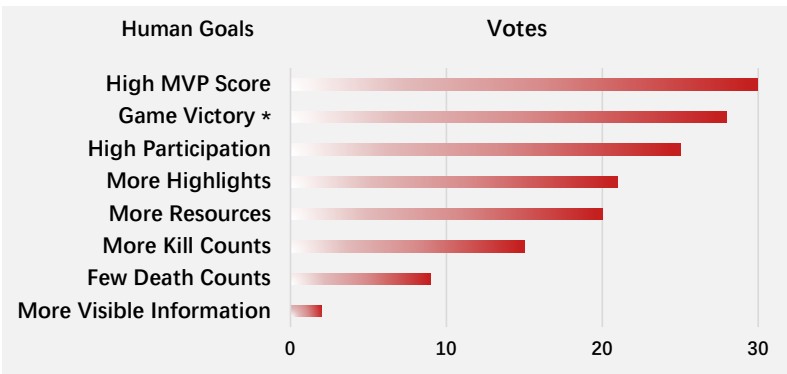

Figure 24: Voting results on human goals in *Honor of Kings*, based on statistics from our participant survey.

### D.2.4 PREFERENCE DESCRIPTION

After each test, participants gave scores on several subjective preference metrics to evaluate their agent teammates, including the **Behavioral Rationality**: the reasonableness of the agent's behavior, the **Enhancement Degree**: the degree to which the agent enhances your abilities to achieve your goals, the **Gaming Experience**: your overall gaming experience, and the **Overall Preference**: your overall preference for your agent teammates.

For each metric, we provide a detailed problem description and a description of the reference scale for the score. Participants rated their agent teammates based on how well their subjective feelings matched the descriptions in the test. The different metrics are described as follows:

- For the Behavioral Rationality, "Do you think your agent teammate's behavior is reasonable? Please evaluate Behavioral Rationality according to the following scales."
    1) Terrible: totally unreasonable.
    2) Poor: more unreasonable behavior.
    3) Normal: some behavior is unreasonable.
    4) Good: less unreasonable behavior.
    5) Perfect: no unreasonable behavior.
- For the Enhancement Degree, "To what extent do you think the agent enhances your abilities to achieve your individual goals? Please evaluate the Enhancement Degree according to the following scales."

   1) Terrible: no enhancements, great reductions.

   2) Poor: no enhancements, slight reductions.

   3) Normal: no enhancements, no reductions.

   4) Good: slight enhancements

   5) Perfect: great enhancements.

- For the Gaming Experience, "How is your gaming experience? Please rate the following words according to your subjective feelings."

   1) Terrible.

   2) Poor.

   3) Normal: close to teaming up with human teammates.

   4) Good.

   5) Perfect.

- For the Overall Preference, "What is your overall preference for your agent teammates? Please rate the following words according to your subjective feelings.".

   1) Terrible.

   2) Poor.

   3) Normal: close to teaming up with human teammates.

   4) Good.

   5) Perfect.

Table 8: The subjective preference results (95% confidence intervals) of all participants in the Human-Agent Game Tests.

| Participant Preference Metrics (from terrible to perfect, 1∼5) | Participant Level | Type of Agent | |
|---|---|---|---|
| | | Wukong | RLHG |
| Behavioral Rationality | General-level | $2.03 \pm 0.31$ | **3.60** $\pm$ **0.30** |
| | High-level | $2.81 \pm 0.21$ | **4.06** $\pm$ **0.30** |
| Enhancement Degree | General-level | $2.10 \pm 0.40$ | **3.50** $\pm$ **0.24** |
| | High-level | $2.94 \pm 0.21$ | **4.14** $\pm$ **0.25** |
| Gaming Experience | General-level | $2.40 \pm 0.41$ | **4.01** $\pm$ **0.30** |
| | High-level | $3.06 \pm 0.27$ | **4.22** $\pm$ **0.30** |
| Overall Preference | General-level | $2.65 \pm 0.35$ | **3.95** $\pm$ **0.28** |
| | High-level | $3.06 \pm 0.21$ | **4.19** $\pm$ **0.29** |

### D.2.5 ADDITIONAL SUBJECTIVE PREFERENCE RESULTS

Detailed subjective preference statistics are presented in Table 8. We can see that both high-level and general-level participants preferred the RLHG agent over the Wukong agent.

**Behavioral Rationality.** We can see that the Behavioral Rationality of the Wukong agent was lower than normal, indicating that participants believed that most of the behaviors of the Wukong agent lacked rationality. The participants generally believed that the behavior of the RLHG agent was more reasonable, therefore they scored the RLHG agent more than normal.

**Enhancement Degree.** Participants believed that the Wukong agent did not bring them any effective enhancement, while they believed that the RLHG agent effectively enhanced their abilities to achieve their individual goals.

**Gaming Experience.** Participants agreed that effective enhancement gave them a good gaming experience, while the irrational behavior of the Wukong agent degraded their gaming experience.

**Overall Preference.** In general, participants were satisfied with the RLHG agent and gave higher scores in the Overall Preference metric. The results of these subjective preference metrics are also consistent with the results of objective performance metrics, further verifying the effectiveness of the RLHG approach.

### D.2.6 Participant Comments

After each game test, participants provided voluntary feedback on their agent teammates. Some participants commented on the RLHG agent "Teaming up with the agent (RLHG) as teammates makes me feel good, they helped me achieve a higher MVP score" and "The agent teammates (RLHG) proactively considered my in-game needs, assisted me in building advantages, and provided the resources I required". Other participants provided feedback on the Wukong agent, stating that "The agent (Wukong) brought me a less enjoyable experience, as they rarely paid attention to my gameplay behavior" and "My agent teammates (Wukong) frequently left me feeling isolated and undervalued". Such voluntary feedback from participants can offer insights into the effectiveness of the RLHG approach.

## E    Broader Impacts

The main goal of our research is to develop better technologies that enable artificial agents to assist humans more effectively in complex environments. This technology has the potential to benefit the research community and various real-world applications, such as friendly assistive robots.

**To the research community.** Games, as the microcosm of real-world problems, have been widely used as testbeds to evaluate the performance of Artificial Intelligence (AI) techniques for decades. And MOBA poses a great challenge to the AI community, especially in the field of Human-Agent Collaboration (HAC). Even though the existing MOBA-game AI systems have achieved or even exceeded human-level performance, they mainly focus on how to compete rather than how to assist humans, leaving HAC in complex environments still to be investigated. To this end, this paper introduces a learning methodology to train agents to assist humans and enhance humans' ability to achieve goals in complex human-agent teaming environments. We herewith expect that this work can provide inspiration for the human enhancement and human assistance in various AI research.

**To the real-world applications.** Firstly, our AI has found real-world applications in games and is changing the way MOBA game designers work. For example, for PVE (player vs environment) teaching mode, introducing AI with human enhancement into the game is a low-cost method to increase the interest of novice players. Secondly, our method can be directly applied to any pre-trained agent, and only needs to be fine-tuned with human gain to change it from apathetic to human-enhanced. It could be directly applied to assistive robotics, such as enhancing the safety of humans in collaboration with industrial robotic arms.

However, we should take into consideration the possibility of human goals being harmful. Therefore, if agents are optimized for harmful goals, this can have negative social impacts, as with all advanced AI techniques, such as AlphaStar (Vinyals et al., 2019), OpenAI Five (OpenAI et al., 2019) and Cicero (FAIR et al., 2022). To avoid these problems, we increase regulation and scrutiny during technological research and development to ensure that human goals do not negatively impact society. In addition, we recommend that when releasing the pre-trained agent model, some restrictions need to be added for fine-tuning, such as enhancing the safety of humans.

## F    Future Work

In some application scenarios, human experience may be difficult to quantify directly but can be implicitly modeled through human feedback, such as in Reinforcement Learning from Human Feedback (RLHF) (Christiano et al., 2017; Ibarz et al., 2018; Ouyang et al., 2022). Therefore, combining RLHF with RLHG for more general collaborative tasks is a promising research direction. Besides, our approach and experiments only consider the scenario where multiple agents enhance one human. Another worthy research direction is how to simultaneously enhance the experience of multiple humans with diverse behaviors.

