# OpenReview forum: "Enhancing Human Experience in Human-Agent Collaboration: A Human-Centered Modeling Approach Based on Positive Human Gain"
_ICLR.cc/2024/Conference — ICLR 2024 poster_

### Official Review · Reviewer_qZtz · 2023-11-03

**Soundness:** 2 fair
**Presentation:** 1 poor
**Contribution:** 3 good
**Rating:** 6
**Confidence:** 4

**Summary:**

This work focusses on fine-tuning high-performing AI agents trained for commercial games to be more enjoyable as teammates for human players.  It has been observed that strong AI policies for team vs. team games can perform well as members of human-AI teams, but nonetheless be unenjoyable for human players due to a lack of perceived cooperation, or opportunities for the human to lead.  They propose to address this issue using "RL from Human Gain" or RLHG, a framework in which the AI's objective is to maximize the impact of the human's own actions on their reward (which may differ from the task-level reward).  They present a policy gradient update that maximizes a mixture of the one-step RLHG objective with the task-level reward.

They evaluate RLHG by fine-tuning the existing Wukong agent developed for the game "Honor of Kings".  Through a user-study with expert players they show that these players generally preferred to play on teams with RLHG-tuned agents rather than the high-performing Wukong agent.

**Strengths:**

I see two main contributions from this paper:
1. Highlighting the importance, in the use of RL for game AI, of maximizing human agency and enjoyment, rather than absolute performance
2. The RL from human gain algorithm for learning policies that maximize human agency itself, rather than

While conceptually simple, they are able to demonstrate empirically that the RLHG algorithm can fine-tune high-performing models into ones that achieve high player satisfaction, while retaining much of their original performance.

**Weaknesses:**

My main concern with this work is that its technical presentation is currently unclear to the point that it is not possible to determine which algorithm is actually being evaluated, yielding two very different possible explanations for the empirical results.

The way RLHG is currently presented suggests that the algorithm attempts to select AI actions that maximize the difference between the advantage of the human's own actions under the AI's action, and their advantage under some baseline AI policy (this advantage captured by the term $\hat{A}_H (s, a)$).  Working through the derivation of the RLHG update, however, it appears that RLHG may essentially be equivalent to optimizing the weighted combination of the human and task rewards $R + \alpha R^H$.  This may simply be a misunderstanding of the notation used, but in either event the presentation needs to be improved for a final version of the paper.

Looking at the RLHG policy gradient in equation 4, it appears that we can express the human gain advantage as:

$$
     \hat{A}_H (s, a) = E_{\pi^H} [ G^{H}_t \vert s_t = s, s_t = a ] - V^{\pi^{e}_{\theta}, \pi^{H}}_H
$$

This is where the notation makes things unclear.  As I understand it, $V^{\pi^{e}_{\theta}, \pi^{H}}_H$ is to be interpreted as a function of the state $s$ alone, in which case it would have no impact on the expectation of the policy gradient (though it would potentially reduce the variance).

The other interpretation, which is more consistent with the intuition behind the algorithm (and with equation 5), is that $V^{\pi^{e}_{\theta}, \pi^{H}}_H$ is really a function of both $s$ and the AI's action $a$ (or equivalently, $V : S \mapsto \Re^A$).

Some improvements to the presentation that could help clear things up:
1. Specify the domain and range of the value functions in question
2. Always condition value functions on their inputs ($V(s)$ instead of $V$)
3. clearly distinguishing in the definitions between the AI actions and those taken by the human

This also raises another question about the relative importance of the "human-gain" objective vs. the distinct human reward function $R^H$ in the empirical results.  My understanding is that the baseline HRE fine-tuning process corresponds to simply maximizing the human's reward function, starting from the pre-trained Wukong agent.  The fact that this performs so poorly would suggest that the human-gain objective is really the critical factor.  Unfortunately the HRE agent is not sufficiently well-defined.  Explicitly writing down the gradient update or objective function for HRE agent would further improve the reader's understanding of RLHG.

If these issue could be clarified it would significantly impact my recommendation.

**Questions:**

1. Were an experiments conducted using RLHG with the task reward and human reward being equivalent (solely optimizing human agency)?
2. Do the HRE experiments correspond to maximizing the $\alpha$-weighted mixture of human and task reward?
3. Was the human-centric reward a pre-existing score in the game environment, or was it derived based on the survey of player preferences?

---

> ### Author Response · Authors · 2023-11-18
> **Response to Reviewer qZtz**
>
> Thank you for carefully reviewing our paper! We greatly appreciate your recognition and positive evaluation of our work. We have carefully revised the manuscript according to your suggestions and have rephrased the parts that caused confusion. If you have any other questions or comments, we are more than happy to discuss them further.
>
> **Q1**: Regarding the technical presentation.
>
> **A1**: We have carefully revised the notations in the manuscript and rewritten Sec. 2.2, Sec. 3.1 and Sec. 3.2 to make the technical presentation clearer. All modifications we marked in red in the rebuttal version. We briefly clarify your confusion below.
>
> Our work aims to train agents to enhance the human experience while maintaining the agent's original abilities as much as possible. Therefore, our approach is essentially optimizing both $R$ and $R^H$, i.e., $J(\theta) = E_{\pi_{\theta},\pi_H} [G_t + \alpha \cdot G_t^H|s_t=s]$, corresponding to the self-centered optimization objective and the human-centered optimization objective, respectively, with the two objectives weighted by $\alpha$.
>
> During the optimization process, we found that the human-centered advantage $A_H(s, a) = E_{\pi_\theta, \pi_H}[G^H_t|s_t=s,a_t=a] - V_H^{\pi_{\theta}, \pi_H}(s)$ in the policy gradient approximation Eq.3 may result in human-agent credit assignment issues. For instance, rewarding agents for assistance when humans can effortlessly achieve their goals is unnecessary, as this could potentially lead to agents learning incorrect behavior and even losing their autonomy (see experimental results in Sec.4.2).
>
> Our key insight to address this challenge is that the human contribution, termed **human primitive value**, i.e., $V_H^{\pi_0, \pi_H}(s)$, should be distinguished from human rewards. $V_H^{\pi_0, \pi_H}(s)$ represents the expected human return for a given state $s$ under the setting of collaboration between the human $\pi_H$ and the pre-trained agent $\pi_0$. The residual benefits, representing the agent's actual contribution in enhancing human to achieve goals, are referred to as **human gains**, i.e., $\Delta(s, a)=E_{\pi_\theta, \pi_H}[G^H_t|s_t=s,a_t=a] - V_H^{\pi_0, \pi_H}(s)$. The agent is solely encouraged to learn effective enhancement behaviors, i.e., $A^{+}(s) = ${$a|a \sim \pi_\theta, \Delta(s, a) > 0$}. We propose the RLHG algorithm to actualize our insight. The RLHG algorithm initially learns a human primitive value network $V_{\phi}$ in the Human Primitive Value Estimation stage. In the subsequent Human Enhancement Training stage, the RLHG algorithm encourages the agent to learn effective enhancement behaviors (actions contributing to positive human gains). For those invalid enhancement behaviors, i.e., $A^{-}(s)=A(s) \setminus A^{+}(s) = ${$a|a \sim \pi_\theta, \Delta(s, a) \le 0$}, the human-centered advantage in Eq.4 is set to 0 and only the self-centered advantage exists to guide the agent's optimization.
>
> **Q2**: Regarding other presentations.
>
> > "Specify the domain and range of the value functions in question".
>
> > "Always condition value functions on their inputs $V(s)$ instead of $V$".
>
> > "Clearly distinguishing in the definitions between the AI actions and those taken by the human".
>
> **A2**: Thank you for your suggestions, we have carefully revised the manuscript according to your suggestions in the rebuttal revision.
>
> **Q3**: Regarding the description of the HRE.
>
> **A3**: HRE (Human Reward Enhancement) essentially integrates the human reward directly into the policy gradient, which is Eq. 3. Based on your suggestion, we marked the gradient update of HRE when introducing the HRE agent in the experiment part, and we also supplemented the detailed optimization process of WuKong and HRE in Appendix C.1.
>
> **Q4**: "Were an experiments conducted using RLHG with the task reward and human reward being equivalent (solely optimizing human agency)?"
>
> **A4**: We conducted experiments to examine the influence of the balance parameter $\alpha$ ($\alpha$ is set to 0, 1, 2, and 4). Please see Appendix C.2 for detailed experimental results.
>
> **Q5**: "Do the HRE experiments correspond to maximizing the $\alpha$-weighted mixture of human and task reward?"
>
> **A5**: Yes, the optimization objective of HRE is the same as RLHG, i.e., $J(\theta) = E_{\pi_{\theta},\pi_H} [G_t + \alpha \cdot G_t^H|s_t=s]$. And the difference between them is the policy gradient update (HRE: Eq.3, RLHG: Eq.4).
>
> **Q6**: "Was the human-centric reward a pre-existing score in the game environment, or was it derived based on the survey of player preferences?"
>
> **A6**: Human goals were obtained through participant survey statistics (Figure 4(c)). These human goals can be quantified as human rewards through pre-existing scores in-game, such as MVP score, highlight moment times, etc.

---

> > ### Comment · Reviewer_qZtz · 2023-11-22
> > **Response to rebuttals**
> >
> > While I believe the authors have addressed my primary concern, and shown that there method is not equivalent to fine-tuning with a different ("human centric") reward function, I feel now that their proposed method may have been mischaracterised.
> >
> > Looking at their revised presentation it seems to me that they real benefit of RL from human gain is that it is resistant to a form of "catastrophic forgetting" when switching from the task reward to the human centric reward.  Looking at Equation 4, it seems that RLHG prevents the fine-tuning process from optimising for the human centric reward in states where the current joint policy is performing worse than the task-specific baseline (the same human policy paired with the original Wukong agent in this case).  This would seem to prevent fine-tuning from forgetting how to perform tasks that are relevant to both the task and human-centric objectives.
> >
> > It is still not clear that the human gain objective captures the extent to which the AI is "empowering" [1] the human to achieve their own goals, as the paper seems to suggest.
> >
> > 1. Du, Yuqing, et al. "Ave: Assistance via empowerment." Advances in Neural Information Processing Systems 33 (2020): 4560-4571.

---

> > > ### Author Response · Authors · 2023-11-22
> > > **Response to Reviewer qZtz**
> > >
> > > We greatly appreciate your feedback and we have promptly revised Eq. 4 accordingly:
> > >
> > > $\nabla J(\theta) = E{\pi\theta, \pi_H} [\sum_{t=0}^{\infty} \nabla_\theta \log \pi_{\theta}(a_t|o_t) (A(s_t,a_t) + \alpha \cdot \widehat{A}_H(s_t, a_t))]$.
> > >
> > > Essentially, RLHG optimizes both task rewards and human rewards simultaneously. We would like to clarify that during the fine-tuning process, RLHG optimizes both self-centered and human-centered objectives simultaneously for all agent behavior samples. However, RLHG only encourages behaviors with human gains higher than the expected positive human gains; for other behaviors (human gains lower than expected positive human gains), the human-centered optimization objective imposes penalties. The difference with HRE is that it encourages behaviors with human returns higher than the expected human-centered value, which may be lower than the human primitive value, resulting to many invaild enhancement behaviors being encouraged to learn.
> > >
> > >
> > > We hope our response can eliminate your confusion and concerns. If you have any other questions or comments, we would be more than happy to reply as soon as possible before the discussion time ends.

---

### Official Review · Reviewer_uNSX · 2023-11-03

**Soundness:** 3 good
**Presentation:** 2 fair
**Contribution:** 4 excellent
**Rating:** 6
**Confidence:** 2

**Summary:**

* This paper proposes a very interesting problem in multi-agent RL that designing robot policies with the goal of providing the human a good experience. The paper proposes a new RL training procedure to obtain such policies and deployed them in the game of "Honor of Kings", in both simulation (4 robot + 1 simulated human playing against 4 robot + 1 simulated human), and in real game play (4 robot + 1 real human playing against 4 robot + 1 simulated human).

* This paper separate the task reward from the human reward, which captures the human experience in the game. Both rewards are known to the system. The training procedure optimizes the robot policies to maximize the difference between the total human reward obtained compared to the total human reward obtained if the robot is just maximizing the task reward.

**Strengths:**

* The formulation and algorithm are sound.
* The result is rich. In both simulation and real game play, the authors demonstrate that the proposed algorithm achieved higher human experience, without sacrificing a lot of task performance.

**Weaknesses:**

* I have some questions about the notations. I think the notation in Sec.2.2 and Sec.3.1, 3.2 are a bit confusing and hard to follow. See my questions in the "questions" box.
* I think Dec-POMDP usually assumes that all agents maximize the same reward function. However, in this paper, the human and robot are not just optimizing the same reward function. Instead, it seems that the human is just a fixed policy that is not optimizing anything. The robot is a policy that optimizes the human gains ∆πH in Sec.3.2. Maybe it is worthy to clarify this.
    * Due to this conflict, it did confuse me in the beginning and took me some time to figure out that in this paper, the human policy is fixed (e.g., learned from data), while the robot policy is learned as the "best-response" to the human policy. In other words, the human might not adapt to or might not be the best respond to the robot policy. It might be useful to make it clear.

**Questions:**

* I think in multi-agent setting, the value function depends on both the robot policies, denoted by π or πθ, in this paper, and also the human policy πH. However, some notations in Sec.2.2 and Sec.3.1, 3.2 are only explicitly associated with one policy, rather than both the robot and human policies, which makes me confused.
    * E.g., in the 2nd line of the paragraph before Eq.1, it says V^πθ, where I am confused that what is the human policy inducing this value function.
    * E.g., in Eq.1's last term - expectation under πH, the total return Gt should depend on the robot policy and the human policy, but the expectation is only taken under πH, not (πH,πθ).
        * Similar to Eq.2's last term - E_πH[...].
        * Similar to the line below Eq.3: E_πH[...].
        * Simlar to Sec.3.2's 2nd paragraph's ∆π(s, a)=E_πH[...].
* I am also not sure how to get the last inequality in page number 4.

---

> ### Author Response · Authors · 2023-11-18
> **Response to Reviewer uNSX**
>
> Thank you for carefully reviewing our paper! We greatly appreciate your recognition of the contribution of our work and your positive review. We have responded to your presentation concerns and carefully revised the manuscript. If you have any further questions or comments, we will be happy to discuss them further.
>
> **Q1**: Questions about the notations.
>
> **A1**: We have carefully revised the notations in the manuscript and rewritten Sec. 2.2, Sec. 3.1 and Sec. 3.2 to make the technical presentation clearer. All modifications we marked in red in the rebuttal version. Below we provide clarification on the notation-related questions you raised.
>
> > **Q1.1**：Regarding "$V^{\pi_{\theta}}(s)$ in the 2nd paragraph of Sec.2.2".
>
> **A1.1**：In the 2nd paragraph of Sec 2.2, we initially stated, "In agent-only team setting",  which does not include humans. In this setting, the optimization objective of the agent is to maximize the cumulative task reward for a given state $s$.  To eliminate possible ambiguities, we have rephrased this paragraph in the rebuttal revision as follows: "Most previous research work focuses on learning self-centered agents. In agent-only team settings, the agent is optimized to complete the task, with the optimization objective being to maximize the value function for a given state $s$, i.e., $V^{\pi_{\theta}}(s) = E_{\pi_{\theta}}[G_t|s_t=s]$, where $G_t=\sum_{k=0}^{\infty}\gamma^{k}R_{t+k+1}$ represents the discounted cumulative task rewards."
>
> > **Q1.2**：Regarding "the total return Gt should depend on the robot policy and the human policy".
>
> **A1.2**: Thank you for pointing this out, we have revised all equations in the manuscript by replacing  $E_{\pi_H}$ with $E_{\pi_{\theta},\pi_H}$.
>
> > **Q1.3**：Regarding "how to get the last inequality in page number 4".
>
> **A1.3**：We have rewritten Sec. 3.2 to make our insights more readable and understandable. After modification, we removed the complex notation representation as well as the inequality.
>
> **Q2**: Clarification about the human policy.
>
> **A2**: We have rephrased the 2nd paragraph of Sec 2.2 and supplemented the description of the human policy to make it clear. "We define a human policy $\pi_H$, to be a mapping from observations $o^H \in \mathcal{O}^H$ to actions $a^H \in \mathcal{A}^H$, which remains fixed during the agent's optimization process and cannot be accessible to the agent."

---

### Official Review · Reviewer_zdWR · 2023-11-04

**Soundness:** 3 good
**Presentation:** 3 good
**Contribution:** 3 good
**Rating:** 6
**Confidence:** 4

**Summary:**

This paper proposes a human-centered modeling scheme to improve the design of human experiments in cooperative human-AI games. Specifically, the authors train two networks: a value network to estimate the expected human return in achieving human goals and a gain network to estimate the expected positive gain of human return. Empirically, they evaluate their proposed method in the game Honor of Kings, and find that the RLHG agent provides participants with a better gaming experience, both in terms of objective performance and subjective preference.

**Strengths:**

The paper's lies in its focus on human experiments in human-AI collaborative games, rather than the current popular AI-centered training scheme. This is a very important area of research, as human experience is essential in human-AI collaboration. Additionally, the paper conducts a comprehensive experimental evaluation of the proposed algorithms, and shows that they can provide a better human experience in terms of both objective performance and subjective preference.

**Weaknesses:**

Although the reviewer agrees that human experience is important in human-agent collaboration, it seems the novelty of this paper is limited. The proposed approach seems to simply add a human modeling network to the existing RL framework.
The details of the algorithms are also not very clear in the paper, and the reviewers suggest that Algorithm 1 should be moved to the main body of the paper to make it easier for readers to follow.

**Questions:**

As weaknesses.

---

> ### Author Response · Authors · 2023-11-18
> **Response to Reviewer zdWR**
>
> Thank you for carefully reviewing our paper! We have responded to your concerns and made revisions to the manuscript based on your valuable suggestions. If you have any further questions or comments, we would be more than happy to discuss them.
>
> **Q1**: Clarification on the novelty.
>
> **A1**: Thank you for your suggestions. We have rewritten Sec. 2.2, Sec. 3.1 and Sec. 3.2 to make the technical presentation clearer and moved a concise version of the algorithm to Sec.3.3 in the rebuttal revision, making it easier for readers to follow our method.
> Our contributions are not limited to network-level modifications in practical implementation, but also include:
>
> - We proposed a human-centered modeling scheme for guiding human-level agents to enhance the experience of their human partners in collaborative tasks.
> - We address the human-agent credit assignment challenge by presenting the RLHG algorithm, along with a detailed implementation framework.
> - We conducted human-agent tests in Honor of Kings, and both objective performance and subjective preference results show that the RLHG agent provides participants better gaming experience.

---

### Author Response · Authors · 2023-11-22
**General Response**

Dear Area Chair and Reviewers,

We thank all reviewers for carefully reviewing our paper and providing valuable comments. We also appreciate the recognition from the reviewers, including:
- Reviewers zdWR and qZtz commented the research problem is important and interesting.
- Reviewer uNSX commented the formulation and algorithm are sound.
- All reviewers commented the experimental results are rich and comprehensive.

Based on the reviewers' valuable suggestions, we have carefully revised the manuscript's notations and rewritten Sec. 2.2, Sec. 3.1, and Sec. 3.2 to enhance the paper's readability and comprehensibility. All modifications we marked in red in the [rebuttal version](https://openreview.net/pdf?id=BqEvdOS1Hs).

We summarize the main revisions as follows:
- We supplemented a detailed description of the human policy in Sec. 2.2 (Reviewer uNSX).
- We provide more clear notations and equations used in the paper. (Reviewer uNSX).
- We moved the RLHG algorithm from Appendix B.7 to Sec. 3.3 to make it easier for readers to follow (Reviewer zdW).
- We improve the technical presentation of the proposed method to eliminate possible ambiguities (Reviewer qZtz, uNSX).
- We supplemented a detailed description, including the optimization objective, of the HRE method in Section 4.1 and Appendix C.1 (Reviewer qZtz).

We believe that after incorporating all reviewers' suggestions, the quality of this work has been improved. If you have any further questions or comments, please feel free to share them. We are looking forward to your feedback.

Best regards,

Paper 7798 Authors

---

### Meta-Review · Area_Chair_e6Cv · 2023-12-06

**Metareview:**

All the reviewers are (mildly) positive about the paper and recommend acceptance. I request the authors to take into account the reviewers' comments and suggestion when preparing the final version of their paper.

**Justification For Why Not Higher Score:**

All the reviewers are mildly positive about the paper but none of them is strongly positive.

**Justification For Why Not Lower Score:**

None of the reviewers is negative about the paper.

---

### Decision · Program_Chairs · 2024-01-16

Accept (poster)